# Unraveling the mechanisms of deep-brain stimulation of the internal capsule in a mouse model

Bastijn J. G. van den Boom [1,2] ✉, Alfredo Elhazaz-Fernandez [1], Peter A. Rasmussen[1], Enny H. van Beest [1], Aishwarya Parthasarathy [1,2], Damiaan Denys[2] & Ingo Willuhn [1,2] ✉

Deep-brain stimulation (DBS) is an effective treatment for patients suffering from otherwise therapy-resistant psychiatric disorders, including obsessive-compulsive disorder. Modulation of cortico-striatal circuits has been suggested as a mechanism of action. To gain mechanistic insight, we monitored neuronal activity in cortico-striatal regions in a mouse model for compulsive behavior, while systematically varying clinically-relevant parameters of internal-capsule DBS. DBS showed dose-dependent effects on both brain and behavior: An increasing, yet balanced, number of excited and inhibited neurons was recruited, scattered throughout cortico-striatal regions, while excessive grooming decreased. Such neuronal recruitment did not alter basic brain function such as resting-state activity, and only occurred in awake animals, indicating a dependency on network activity. In addition to these widespread effects, we observed specific involvement of the medial orbitofrontal cortex in therapeutic outcomes, which was corroborated by optogenetic stimulation. Together, our findings provide mechanistic insight into how DBS exerts its therapeutic effects on compulsive behaviors.

Electrical deep-brain stimulation (DBS) is used to treat a growing number of neurological and psychiatric disorders[1]. In psychiatry, the most common stimulation target is the anterior limb of the internal capsule (IC)[2], a major fiber bundle that carries axons connecting the cortex with the striatum and other subcortical structures[3,4]. For example, patients suffering from obsessive-compulsive disorder (OCD), a disorder characterized by unwanted thoughts (obsessions) and repetitive behaviors (compulsions)[5,6], that are resistant to conventional therapy benefit substantially from DBS[7–9]. However, despite decades of the clinical application of DBS, the precise neurobiological mechanisms underlying its therapeutic success remain insufficiently understood. This insufficient understanding is reflected in limited therapeutic effect size and long periods of trial-and-error DBS-parameter optimization in OCD patients and other psychiatric disorders[10,11].

A historically influential theory postulates that DBS inhibits neural tissue surrounding the electrode tip ("inhibition hypothesis") and thereby creates a reversible lesion with clinical efficacy similar to irreversible lesions of the subthalamic nucleus in neurological movement disorders[12] and to that of capsulotomy in psychiatry[13]. More recently, a modified version of this idea has been favored: DBS is assumed to act as a (reversible) "information lesion"[14,15], potentially by interfering with the ability of neurons to properly respond to synaptic input[15]. Thus, instead of exclusively (and completely) inhibiting neuronal activity, as assumed by the inhibition hypothesis, DBS is thought to create a disturbance on an "information level", which may promote both excitation and inhibition in the vicinity of the electrode. This is supported by findings that demonstrate that DBS can cause both excitation and inhibition of neurons locally[16–18].

[1]Netherlands Institute for Neuroscience, Royal Netherlands Academy of Arts and Sciences, Amsterdam, The Netherlands. [2]Department of Psychiatry, Amsterdam UMC, University of Amsterdam, Amsterdam, The Netherlands. ✉e-mail: bastijnvandenboom@gmail.com; i.willuhn@nin.knaw.nl

On the brain systems level, several studies point to dysfunctional activity in cortico-striatal circuits in OCD[19–24], which is thought to be corrected by DBS via the recruitment of cortical regions[2,4,25–29]. For example, DBS has been reported to decrease low-frequency oscillations in the prefrontal cortex (PFC) that had been induced by OCD-symptom provocation[25]. Furthermore, a recent study suggests that DBS improves cognitive control in OCD patients via the modulation of oscillations in the medial and lateral PFC[28]. The orbitofrontal cortex (OFC), another subregion of the PFC, has consistently been shown to exhibit diminished glutamate metabolism in response to DBS[30–32]. Furthermore, dysfunction is generally thought to be characterized by enhanced functional connectivity of PFC regions with downstream striatal regions[22,23,33]. Such excessive connectivity within cortico-striatal circuits can predict symptom severity, where both excessive connectivity and symptoms can be reduced by DBS[34]. Moreover, DBS-induced reduction in functional connectivity within cortico-striatal circuits correlated to the degree of OCD-symptom alleviation[25]. Remarkably, DBS targeting projections from medial and lateral aspects of the OFC to the striatum ameliorated OCD symptoms with nuanced differences, indicating projection specificity[35], where improved therapeutic efficacy can be achieved by specifically targeting cortico-striatal projections that originate from cortical regions that responded to OCD-symptom provocation[36]. Together, these studies indicate a role for cortico-striatal circuits in the pathophysiology of OCD, as well as for the therapeutic effects of DBS in OCD. However, which sub-domains of such cortico-striatal circuits are most important is still under investigation.

To elucidate the mechanisms by which DBS acts in cortical and striatal regions, we employed SAPAP3 mutant mice (SAPAP3[−/−]), a well-established model for OCD. These mice exhibit compulsive-like grooming[37–41], anxiety-like behavior[38,42], cognitive deficits[40,42–44], and respond well to OCD pharmacotherapy[38,45] and DBS[46]. To avoid DBS-induced electrical artifacts, we used calcium imaging to monitor single-cell activity in vivo[47]. We applied DBS to the rodent homolog of the human ventral anterior IC, the mouse ventral IC, which carries similar cortical projection fibers[3]. Across experiments, we systematically varied clinically-relevant DBS parameters (current, pulse width, and frequency) in SAPAP3[−/−] and their wild-type littermates (WT)[10]. We report a DBS dose-dependent reduction in excessive grooming, accompanied by both brain-wide neuronal dynamics and responses specific to the medial OFC.

## Results

### Internal-capsule deep-brain stimulation (IC-DBS) decreases excessive grooming dose-dependently

We examined the effects of DBS parameters on compulsive-like grooming behavior in SAPAP3[−/−] ($n = 30$) and WT ($n = 28$) (Supplementary Fig. 1a–c). Animals were tested in an open-field apparatus after implanting DBS electrodes into the IC (Fig. 1a–c, Supplementary Fig. 1d). All mice were stimulated across three different experiments (on different days) to examine the effects of different intensities (no DBS, low-, medium-, and high-intensity DBS) of clinically relevant DBS parameters (current, pulse width, and frequency) on brain and behavior (Fig. 1d, e). Consistent across experiments, SAPAP3[−/−] spent approximately 20% of the open-field session grooming at baseline (no DBS), whereas WT groomed only for 5% (Fig. 1f, h). For each of the three DBS-parameter experiments, SAPAP3[−/−] grooming was not reduced during the lowest DBS-intensity condition (pre-DBS baseline vs DBS; green, 100 µA: t(26) = 1.78, p = 0.087; 40 µs: t(27) = 0.52, p = 0.608; 60 Hz: t(25) = 1.31, p = 0.203). However, both medium- (yellow) and high-intensity (red) DBS conditions showed immediate and robust reductions in excessive grooming (pre-DBS baseline vs DBS; 200 µA: t(26) = 2.34, p = 0.040; 300 µA: t(26) = 4.44, p = 0.003; 80 µs: t(27) = 2.35, p = 0.039; 160 µs:

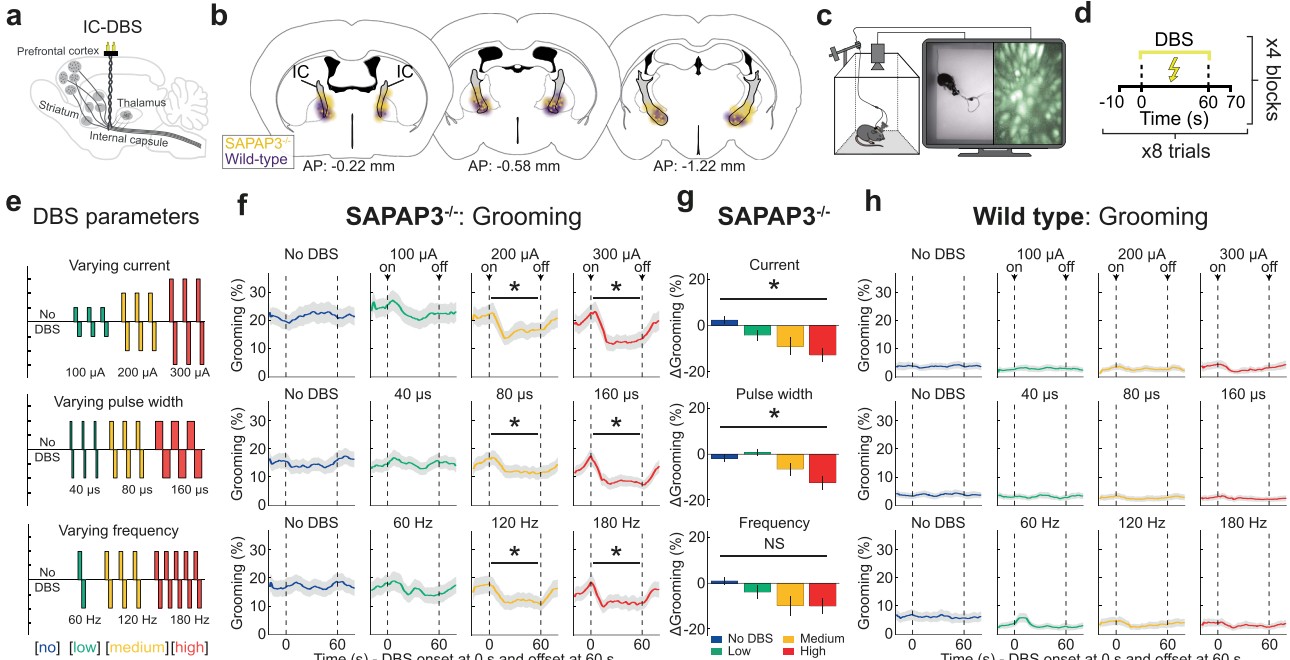

**Fig. 1 | Internal-capsule deep-brain stimulation (IC-DBS) dose-dependently reduces excessive grooming. a** Schematic depicting DBS electrodes in the IC, a white-matter bundle that carries corticofugal fibers. **b** Histological verification of IC-DBS electrode tips in the IC (gray) of SAPAP3[−/−] (yellow, n = 30) and WT (purple, n = 28) projected on the Allen Reference Atlas - Mouse Brain[101]. Halo represents the modeled sphere of current spread around the DBS electrode tips. **c** Mice were subjected to DBS, calcium imaging, and behavioral recordings after being placed in an open field. **d** DBS was switched ON for 60 s per trial, eight trials per block, and four blocks per session. **e** Across three sessions, animals were stimulated with varying current (top), pulse width (middle), or frequency (bottom). **f** DBS reduced compulsive-like grooming during current (top), pulse width (middle), and frequency (bottom) dose-response experiments. **g** A significant dose-dependent reduction of grooming was observed during current (n = 27) and pulse-width (n = 28), but not frequency (n = 26), manipulations. **h** DBS did not reduce grooming in WT mice. Data are mean ± SEM. Source data are provided as a Source Data file. AP anterior-posterior, *p < 0.05, NS not significant.

$t(27) = 4.18$, $p = 0.005$; 120 Hz: $t(25) = 2.27$, $p = 0.048$; 180 Hz: $t(25) = 2.94$, $p = 0.014$) (Fig. 1f). Excessive grooming rapidly reinstated upon DBS offset. Both current and pulse-width experiments exhibited a dose-dependent reduction in grooming (current: $F(3,78) = 7.82$, $p < 0.001$; pulse width: $F(3,81) = 7.02$, $p < 0.001$), whereas increasing frequency beyond 120 Hz did not improve efficacy further (frequency: $F(3,75) = 1.97$, $p = 0.126$) (Fig. 1g). Importantly, DBS did not alter WT grooming (Fig. 1h) or general locomotion (Supplementary Fig. 1f, g). Surprisingly, the precise DBS-electrode location (two-dimensional anterior-posterior and dorsal-ventral position) within the mouse ventral IC did not significantly correlate with grooming reduction (Bayes factors for current: 6.2, pulse width: 3.7, frequency: 4.9) (Supplementary Fig. 1e). Thus, DBS is effective as long as the electrodes are located within the ventral IC, likely due to the sufficiently large sphere of DBS-current distribution (Fig. 1b; yellow and purple halos depict estimated current distribution) across the ventral IC that influences most ventral-IC fibers irrespective of precise electrode coordinates. Exploring novel DBS parameters (low frequency and "cyclic" stimulation) did not result in grooming reduction, in line with recent patient findings[48] (Supplementary Fig. 1h, i). Taken together, IC-DBS reduced excessive grooming in SAPAP3[−/−] and, similar to clinical practice, the effectiveness of DBS was improved by adapting current and pulse width, but not frequency[10].

## IC-DBS modulates the entire dorsal cortex, with an emphasis on the frontal cortex

DBS is thought to recruit cortical regions[4]. To assess widespread cortical effects of IC-DBS, we employed wide-field calcium imaging across the entire dorsal cortex in *Thy1*-GCaMP6f mice ($n = 5$)[49]. The skull was made transparent for calcium imaging[50] and IC-DBS electrodes implanted, targeted at the ipsilateral hemisphere (Fig. 2a, Supplementary Fig. 2a). After motion correction (Supplementary Fig. 2b–d), the Allen-brain atlas was used to map neuronal activity onto specific brain regions[51] (Fig. 2b). Similar to the dose-dependent reduction in

grooming during varying current and pulse-width experiments (Fig. 1g), we found dose-dependent suppression of the entire dorsal cortex (current: $F(3,12) = 4.38$, $p = 0.027$; pulse width: $F(3,12) = 7.20$, $p = 0.005$), which was absent in the frequency experiment (frequency: $F(3,12) = 0.31$, $p = 0.845$) (Fig. 2c). Analyzing brain regions independently, we found increased activity in all recorded regions immediately upon DBS onset (Fig. 2d). However, within a few seconds activity diminished and sustained suppression of activity was detected in a subset of regions. In the frontal cortex (FC), we found suppression across current ($F(3,12) = 8.77$, $p = 0.005$) and pulse-width ($F(3,12) = 9.64$, $p = 0.005$) experiments, and in the somatosensory cortex (SS) only during pulse-width manipulations ($F(3,12) = 6.41$, $p = 0.023$). We observed no significant suppression in retrosplenial cortex (RSP) nor visual cortex (VIS), suggesting a rostral-caudal gradient of suppression (Fig. 2d). Direct comparison between regions revealed prominent suppression in FC during current ($F(3,12) = 4.20$, $p = 0.023$; post-hoc: FC vs RSP $p = 0.043$, FC vs VIS $p = 0.034$) and pulse-width ($F(3,12) = 3.51$, $p = 0.040$; post-hoc: FC vs VIS $p = 0.040$) experiments, but not frequency ($F(3,12) = 2.95$, $p = 0.065$) (Fig. 2e). These data indicate that IC-DBS sustainedly suppresses the entire dorsal cortex with a rostral-to-caudal gradient.

## Single-cell recruitment in cortical and striatal regions by IC-DBS

Wide-field imaging captures a large part of the brain, but lacks single-cell resolution and is limited to superficial cortical layers in head-fixed mice[52]. To overcome these limitations and elucidate DBS effects on grooming in OCD-relevant circuits, we used miniaturized fluorescent microscopes (miniscopes, Fig. 3a)[53]. Consistent with literature, our wide-field data indicated a role for the FC in IC-DBS[25]. Therefore, we used miniscopes to record from putative pyramidal neurons in PFC regions (lateral and medial orbitofrontal cortex (lOFC, mOFC), pre-limbic cortex (PL), and premotor cortex (M2)) and predominantly medium-spiny neurons in their striatal projection targets (dorsal and

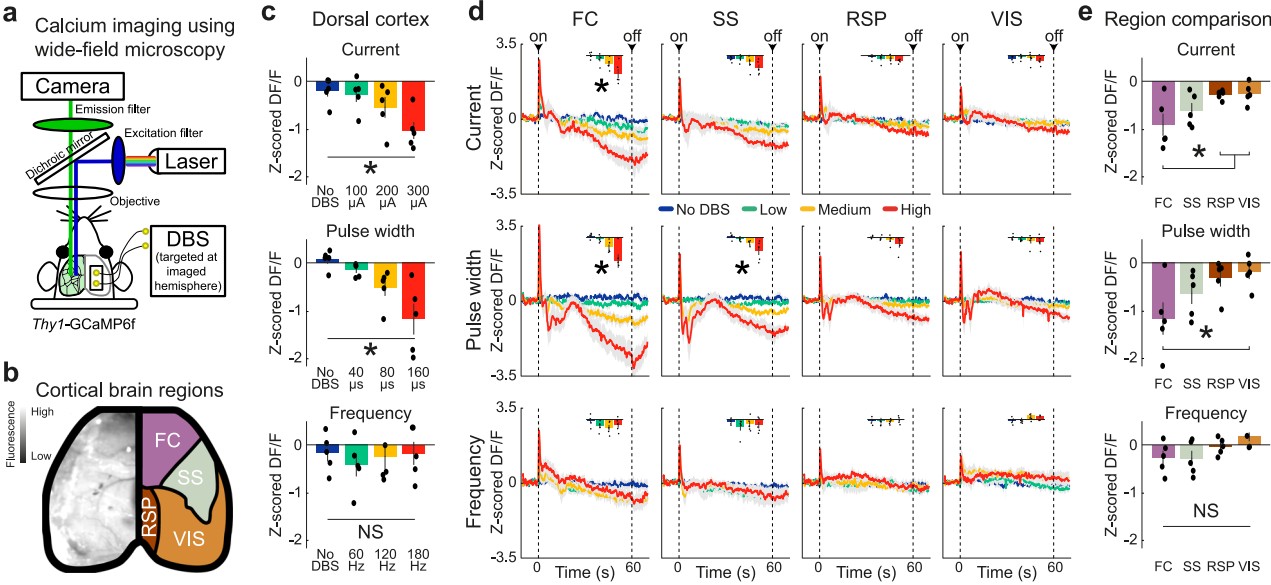

**Fig. 2 | Wide-field calcium imaging reveals IC-DBS modulation of the entire dorsal cortex, with an emphasis on the FC. a** Schematic of wide-field fluorescence microscopy setup. DBS electrodes were implanted, inserted at an angle, contralaterally to the imaged hemisphere, and targeted the ipsilateral IC of *Thy1*-GCaMP6f mice ($n = 5$). **b** A video frame of the dorsal cortex (left) and the cortical regions (FC frontal cortex, RSP retrosplenial cortex, SS somatosensory cortex, VIS visual cortex) as defined by the Allen Mouse Brain Connectivity Atlas (right)[51]. **c** Significant dose-dependent cortex-wide suppression was observed in the varying current ($n = 5$) and pulse-width ($n = 5$) experiments, but not in the frequency ($n = 5$)

experiment. **d** The dose-response manipulations of DBS induced region-specific reduction in FC (current and pulse width) and SS (pulse width), but not RSP nor VIS. **e** Directly comparing sustained suppression at high-intensity DBS across dorsal-cortical regions revealed significant effects during current and pulse-width, but not frequency, experiments. Post-hoc analyses revealed differences between FC and RPS, and FC and VIS during current manipulations. During pulse-width manipulations, FC differed from VIS. Data are mean ± SEM. Source data are provided as a Source Data file. *$p < 0.05$, NS not significant.

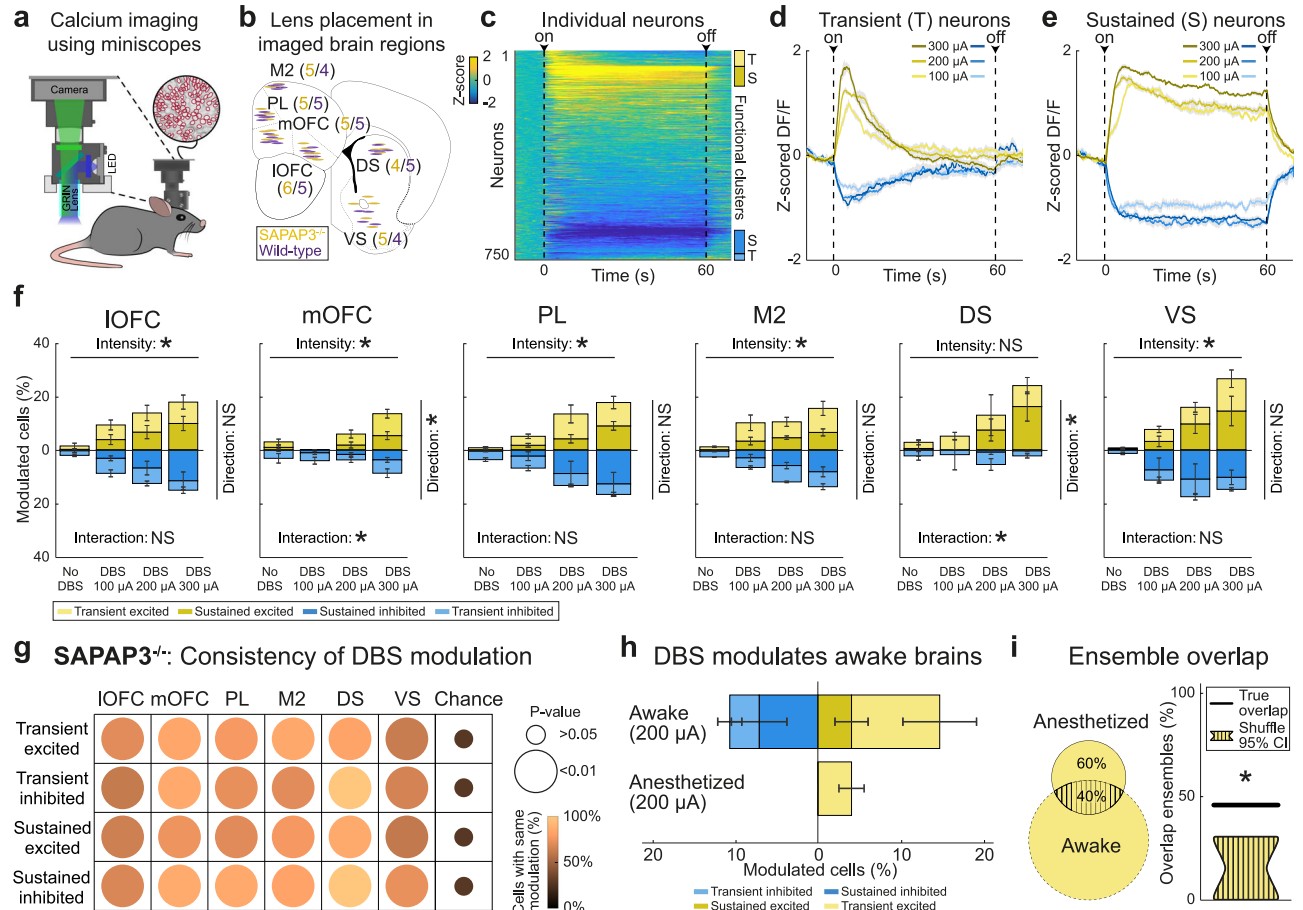

**Fig. 3 | IC-DBS recruits neurons in SAPAP3$^{-/-}$ in cortical and striatal regions via transient/sustained excitation or inhibition. a** Calcium imaging in freely-moving mice using miniscopes. **b** Horizontal elliptic shapes represent imaged areas under GRIN lenses in SAPAP3$^{-/-}$ (yellow, $n = 30$) and WT (purple, $n = 28$) in all recorded regions (lateral and medial orbitofrontal cortex (lOFC, mOFC), prelimbic cortex (PL), secondary motor cortex (M2), and dorsal and ventral striatum (DS, VS)), projected on the Allen Reference Atlas - Mouse Brain[101]. **c** Heatmap of all recorded neurons in lOFC (across all SAPAP3$^{-/-}$ mice) during high-intensity DBS, sorted by modulation. Vertical bars on the right depict functional clusters: transient excited (T, light yellow), sustained excited (S, dark yellow), sustained inhibited (S, dark blue), and transient inhibited (T, light blue). **d** Averaged traces of transiently excited and inhibited neurons during different DBS parameters. **e** Same as **d**, but for sustainedly recruited neurons. **f** Dose-dependent recruitment of excited and

inhibited neurons by DBS in SAPAP3$^{-/-}$ were balanced in number in most regions (lOFC ($n = 6$), PL ($n = 5$), M2 ($n = 5$), and VS ($n = 5$)). However, we found an imbalance in the number of recruited excited and inhibited neurons in mOFC ($n = 5$) and DS ($n = 4$). **g** Stability of direction (excitation or inhibition) and duration (transient or sustained) of single-cell recruitment (i.e., consistency of modulation) was significantly different from chance in all recorded cortical and striatal regions. **h** Under anesthesia, no sustained neurons (either excited or inhibited), nor transient inhibited neurons, were found ($n = 4$). **i** Overlap of identity of transiently excited neurons found between awake and anesthetized states was significantly different from chance (95% confidence intervals presented). Data are mean ± SEM. Source data are provided as a Source Data file. CI=confidence interval, *$p < 0.05$, NS not significant.

ventral striatum (DS, VS)) (Fig. 3b, Supplementary Fig. 3a). During open-field experiments with freely-behaving SAPAP3$^{-/-}$ and WT, we recorded fluorescence in hundreds of neurons per region and found complex dynamics (Fig. 3c, Supplementary Fig. 3b–d). A subset of neurons exhibited immediate, transient effects (either excitatory or inhibitory) upon DBS onset (Fig. 3d, Supplementary Fig. 3e), resembling cortex-wide increases in activity at DBS onset (Fig. 2d). Other neurons showed sustained excitation or inhibition of their activity during DBS (Fig. 3e, Supplementary Fig. 3f), resembling the sustained suppression we found in the FC (Fig. 2d). To describe how DBS modulates neuronal activity, we use the term "recruitment". We define recruitment as a significant change in activity, irrespective of direction of change (increase or decrease), duration of change (transient or sustained), or origin of change (direct entrainment or indirect, network-driven modulation). In all recorded brain regions, DBS recruited single cells dose-dependently by exciting or inhibiting their activity (Factor intensity, lOFC: F(3,40) = 5.53, $p = 0.003$; mOFC: F(3,32) = 12.03, $p < 0.001$; PL: F(3,32) = 11.27, $p < 0.001$; M2:

F(3,32) = 14.54, $p < 0.001$; DS: F(3,24) = 2.57, $p = 0.077$; VS: F(3,32) = 8.75, $p < 0.001$), which occurred similarly in SAPAP3$^{-/-}$ and WT (Fig. 3f, Supplementary Fig. 3h). However, mOFC and DS predominantly recruited neurons by exciting their activity, independent of genotype (Factor direction, mOFC: F(3,32) = 13.67, $p < 0.001$; DS: F(3,24) = 7.68, $p = 0.011$). Neurons that were recruited by DBS showed consistency in duration (transient or sustained) and direction (excited or inhibited) across different DBS intensities in SAPAP3$^{-/-}$ (Fig. 3g) and WT (Supplementary Fig. 3g) (e.g., a sustainedly excited recruited neuron during one DBS intensity is likely to be recruited as sustainedly excited again during another DBS intensity), suggesting that DBS affects neurons similarly across different stimulation parameters. Since behavioral DBS effects were restricted to periods of stimulation, we hypothesized that sustained neurons drive grooming reduction in SAPAP3$^{-/-}$. Supporting this hypothesis, we found a negative correlation between the number of sustained neurons in lOFC and mOFC and grooming reduction (Supplementary Fig. 3i). To investigate whether sustained recruitment is driven by "direct" antidromic stimulation or "indirectly" by network

activity, we imaged mice under both awake and anesthetized (diminished network activity[54]) conditions in the same session. Under anesthesia, we found no sustained neurons (neither excited or inhibited), nor transient inhibited neurons (Fig. 3h), demonstrating that sustained recruitment is dependent on (indirect) network activity in the awake state. Similar to the consistent recruitment of neurons (direction and duration), we found that transient excited neurons recruited under anesthesia were more likely to be recruited as transient excited neurons in awake recordings (bootstrap, $p = 0.002$) (Fig. 3i), suggesting (direct) antidromic stimulation of their axons. Together, DBS modulated neuronal activity in a dynamic, yet consistent fashion that was dependent on network activity.

## Unaffected basic brain function during IC-DBS

To examine if DBS alters all aspects of neuronal activity, we quantified proxies of "basic brain function": We calculated the baseline activity of neurons (a form of resting-state activity)[55] and examined how anatomical distance between neurons affected the synchrony of their activity (spatiotemporal correlations)[56]. First, we compared averaged regional baseline activity during the "no-DBS" block with baseline activity during the high-intensity DBS block. The cumulative density function of calcium events and the average frequency of calcium events were combined into a cell-activity index measure (Fig. 4a). The cell-activity index during the high-intensity DBS block was not different from the "no-DBS" block in SAPAP3$^{-/-}$ (Fig. 4b) or WT

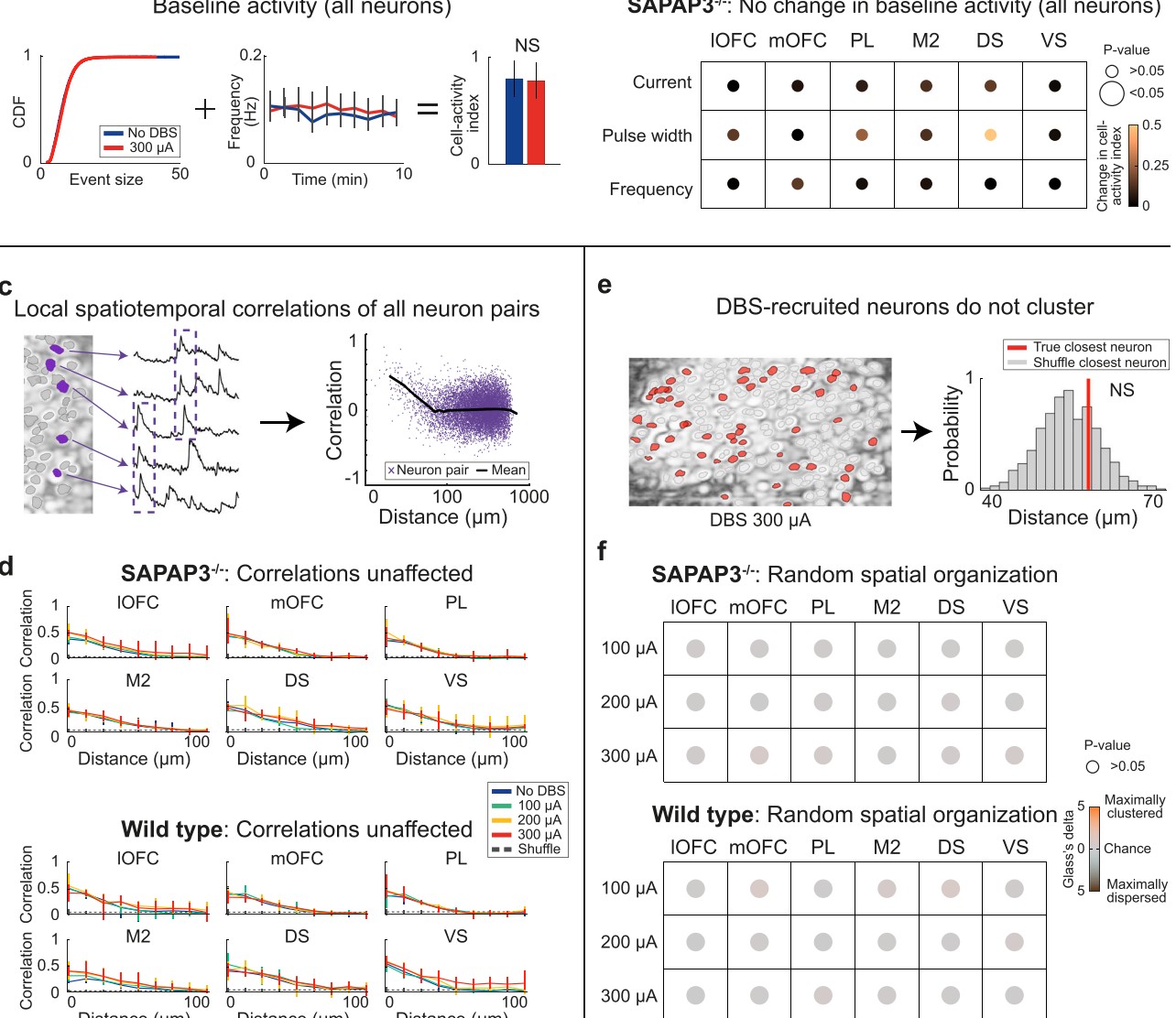

**Fig. 4 | Basic brain function not affected by IC-DBS. a** Cumulative distribution function (left) and averaged frequency of firing (middle) during the "no-DBS" block and the high-intensity DBS block were combined into a single neuronal-activity measure: Cell-activity index (right) ($n = 750$ neurons), which did not differ between the two blocks. **b** High-intensity DBS did not change the cell-activity index compared to the "no-DBS" block in SAPAP3$^{-/-}$. **c** Representative neurons (purple) show synchronous activity when proximal to one another (left). Temporal activity correlation plotted as a function of inter-cell distance (right; purple crosses, individual neuron pairs; black line, average). Distance was logarithmically scaled for visualization. **d** For both SAPAP3$^{-/-}$ and WT, local spatiotemporal correlations were conserved during DBS, as DBS did not differ from the "no-DBS" block (SAPAP3$^{-/-}$: lOFC ($n = 6$), mOFC ($n = 5$), PL ($n = 5$), M2 ($n = 5$), DS ($n = 4$), VS ($n = 5$); WT: lOFC ($n = 5$), mOFC ($n = 5$), PL ($n = 5$), M2 ($n = 4$), DS ($n = 5$), VS ($n = 4$). **e** Maximum-intensity projection displaying the location of recruited neurons highlighted in red (left). Example of averaged distance to closest recruited neuron (red) compared to histogram of expected closest neighbor by chance (bootstrap, gray) (right). **f** No statistically significant clustering for SAPAP3$^{-/-}$ and WT was found in any of the recorded cortical and striatal regions for any of the DBS parameters. Data are mean ± SEM. Source data are provided as a Source Data file. NS not significant.

(Supplementary Fig. 4a), indicating that resting-state activity was not affected by DBS. Next, we tested whether spatiotemporal correlations (i.e., distance between neuron pairs correlated with their activity)[57] were preserved during DBS (Fig. 4c). Local spatiotemporal correlations between recorded neurons were found in all regions during the "no-DBS" block and were preserved during DBS (Fig. 4d). Since DBS strongly modulated neuronal activity (Fig. 3f), we reasoned that preserved local spatiotemporal correlations would likely be achieved by scattered recruitment of neurons (i.e., lack of spatial clustering). To properly assess anatomical organization (Supplementary Fig. 4b), we calculated the anatomical distance of each recruited neuron to its closest recruited neighbor (to avoid averaging out short and long distances between neuron pairs), compared to chance (Fig. 4e), and did not find clustering of neurons (Fig. 4f). Together, these data indicate that, although DBS recruits neurons scattered throughout brain regions, it does not compromise basic brain function.

### Dose-dependent recruitment of a balanced number of excited and inhibited neurons

Next, we focused our analyses on neurons that were recruited sustainedly throughout the stimulation epoch, because their activity change paralleled the duration of grooming reduction, was dependent on network activity (in the awake state), and negatively correlated with grooming reduction. Sustained neurons (both excited and inhibited) were found throughout the miniscope's field-of-view in each DBS-intensity block (Fig. 5a). In all regions, we found dose-dependent recruitment of sustained neurons that amounted to a maximum of approximately a quarter of all imaged neurons in both SAPAP3$^{-/-}$ (lOFC: $F_{(3,15)} = 7.93$, $p = 0.002$; mOFC: $F_{(3,12)} = 15.72$, $p < 0.001$; PL: $F_{(3,12)} = 8.93$, $p = 0.002$; M2: $F_{(3,12)} = 31.37$, $p < 0.001$; DS: $F_{(3,9)} = 4.32$, $p = 0.038$; VS: $F_{(3,12)} = 5.27$, $p = 0.015$) and WT (lOFC: $F_{(3,12)} = 31.24$, $p < 0.001$; mOFC: $F_{(3,12)} = 18.40$, $p < 0.001$; PL: $F_{(3,12)} = 6.59$, $p = 0.006$; M2: $F_{(3,9)} = 5.50$, $p = 0.020$; DS: $F_{(3,12)} = 7.77$, $p = 0.004$; VS: $F_{(3,9)} = 13.13$, $p = 0.001$) (Fig. 5b). Importantly, when comparing SAPAP3$^{-/-}$ with WT, we found stronger recruitment in the mOFC in SAPAP3$^{-/-}$ in each DBS-parameter experiment (current: $F_{(1,16)} = 8.40$, $p = 0.010$; pulse width: $F_{(1,16)} = 9.96$, $p = 0.006$; frequency: $F_{(1,16)} = 7.11$, $p = 0.017$) (Fig. 5c), suggesting that the mOFC in SAPAP3$^{-/-}$ was affected more robustly by DBS than in WT and potentially drove the reduction of excessive grooming. Since neurons have a predisposition to being recruited in a particular manner (Fig. 3g) and DBS recruited overlapping transient-neuron populations in awake and anesthetized mice (Fig. 3i), we tested whether DBS would also recruit overlapping sustained neuron populations across DBS-intensity blocks (Fig. 5d). Neurons recruited during medium- and high-intensity overlapped above chance level in both SAPAP3$^{-/-}$ (bootstrap - lOFC: $p = 0.003$; mOFC: $p = 0.001$; PL: $p = 0.003$; M2: $p = 0.003$; DS: $p = 0.018$; VS: $p = 0.024$) and WT (bootstrap - lOFC: $p = 0.003$; mOFC: $p = 0.018$; PL: $p = 0.003$; M2: $p = 0.005$; DS: $p = 0.003$; VS: $p = 0.003$) (Fig. 5e, Supplementary Fig. 5a, b). These data are consistent with our behavioral results (Fig. 1f): Only medium- and high-intensity DBS reduced excessive grooming. Since neurons can be excited or inhibited by DBS modulation, we investigated whether a computed excitation/inhibition (E/I) balance observed at baseline was maintained during DBS (Fig. 5f). In most regions, we found preserved E/I balance, as demonstrated by similar number of neurons recruited as excited and inhibited (Fig. 5g). However, during DBS, mOFC and DS neurons were predominantly excited in SAPAP3$^{-/-}$ (mOFC: $F_{(3,16)} = 3.68$, $p = 0.035$; DS: $F_{(3,12)} = 6.18$, $p = 0.009$) and WT (mOFC: $F_{(3,16)} = 3.92$, $p = 0.028$; DS: $F_{(3,16)} = 5.33$, $p = 0.010$), providing more evidence for the involvement of the mOFC in DBS effects. Together, these data demonstrate "global" DBS effects (which were independent of genotype and brain region): dose-dependent recruitment of neuron populations that partially overlap at therapeutic intensities, while maintaining E/I balance in most regions. These global effects were accompanied by "regional" DBS effects (which were dependent on brain

region): the mOFC recruitment in SAPAP3$^{-/-}$, primarily by means of excitation, suggests that mOFC potentially drives the DBS-induced reduction of grooming in SAPAP3$^{-/-}$.

### mOFC in SAPAP3$^{-/-}$ controls compulsive-like grooming

Since DBS selectively reduced grooming, we identified neurons that were modulated specifically during grooming in SAPAP3$^{-/-}$ using Bayesian ANOVAs (Bayes factor>3 to obtain substantial evidence for the presence or absence of effects) (Fig. 6a, Supplementary Fig. 6a, b). Even using relatively strict Bayesian identification criteria (see Methods), we identified such neurons in all recorded regions (Fig. 6b). For the sake of simplicity, we refer to them as "grooming-associated" neurons. Similarly, we refer to neurons that were not modulated during periods of grooming, locomotion, or behavioral inactivity as "not-associated" neurons. In the mOFC of SAPAP3$^{-/-}$, we found dose-dependent recruitment of grooming-associated neurons (mOFC: $F_{(3,16)} = 4.13$, $p = 0.024$) (Fig. 6c, top row). In contrast, in all cortical regions, we found dose-dependent recruitment of not-associated neurons (lOFC: $F_{(3,20)} = 4.74$, $p = 0.012$; mOFC: $F_{(3,16)} = 3.31$, $p = 0.047$; PL: $F_{(3,16)} = 8.25$, $p = 0.002$; M2: $F_{(3,16)} = 13.18$, $p < 0.001$) (Fig. 6c, bottom row), suggesting that DBS does not recruit neurons based on their "cell identity", which was supported by a lack of overlap between behavior-associated and DBS-recruited neuron populations (Supplementary Fig. 6c–f). We hypothesized that DBS reduces grooming by diminishing the number of grooming-associated neurons. This hypothesis is supported by the finding that the number of grooming-associated neurons in mOFC was consistently reduced during DBS across DBS-parameter experiments (mOFC - current: $t_{(4)} = 3.98$, $p = 0.016$; pulse width: $t_{(4)} = 2.97$, $p = 0.041$; frequency: $t_{(4)} = 2.98$, $p = 0.041$) (Fig. 6d), suggesting that the activity of mOFC neurons contributes to the expression of grooming. To test whether these neurons causally contribute to grooming reduction, we expressed the excitatory opsin ChETA[58] in mOFC ($n = 7$) to mimic regional excitatory DBS effects (Figs. 5g and 6e, Supplementary Fig. 6g). Indeed, when photostimulation (473 nm, 5 mW, 10 ms pulse-duration) was delivered for 60 s (to mimic the activity of sustainedly recruited neurons), grooming diminished at photostimulation onset and re-emerged at offset (5 Hz: $t_{(6)} = 3.86$, $p = 0.033$; 15 Hz: $t_{(6)} = 4.17$, $p = 0.029$; 120 Hz: $t_{(6)} = 3.58$, $p = 0.035$) (Fig. 6f, g). In contrast, 60 s photostimulation at 1 Hz or photostimulation at 15 Hz for 5 s (to mimic the activity of transiently recruited neurons) had no effect on grooming. Photostimulation, regardless of protocol, did not affect general locomotion or the relationship between grooming and locomotion (Supplementary Fig. 6h, i). Changes in grooming were abolished when light was prevented from entering the mOFC by blocking the implanted fibers (Fig. 6g). Importantly, animals ($n = 5$) expressing only the stable fluorophore mCherry in mOFC and animals expressing ChETA in lOFC ($n = 5$) or M2 ($n = 5$) did not display reduced grooming. Taken together, mOFC neurons exhibit grooming-related information that is altered by DBS and is tightly linked to excessive grooming.

## Discussion

Although DBS is widely used for the treatment of otherwise therapy-resistant OCD patients, its mechanisms of action remain understood insufficiently[11]. In particular, DBS-induced electrical artifacts have hampered the electrophysiological investigation of the brain during DBS[29,59]. Here, we overcame this limitation by using electrical-noise-resistant calcium imaging[47] in freely-moving SAPAP3$^{-/-}$, a well-established mouse model for OCD[38], to monitor cortex-wide population dynamics and single-cell activity in cortical and striatal regions, while systematically varying clinically relevant DBS parameters. We identified several "global" and "regional" effects of IC-DBS. The DBS effects occurring uniformly across genotype and brain region, the "global" effects, were: 1) Both direction (excitation versus inhibition) and duration (transient versus sustained) of the recruitment of

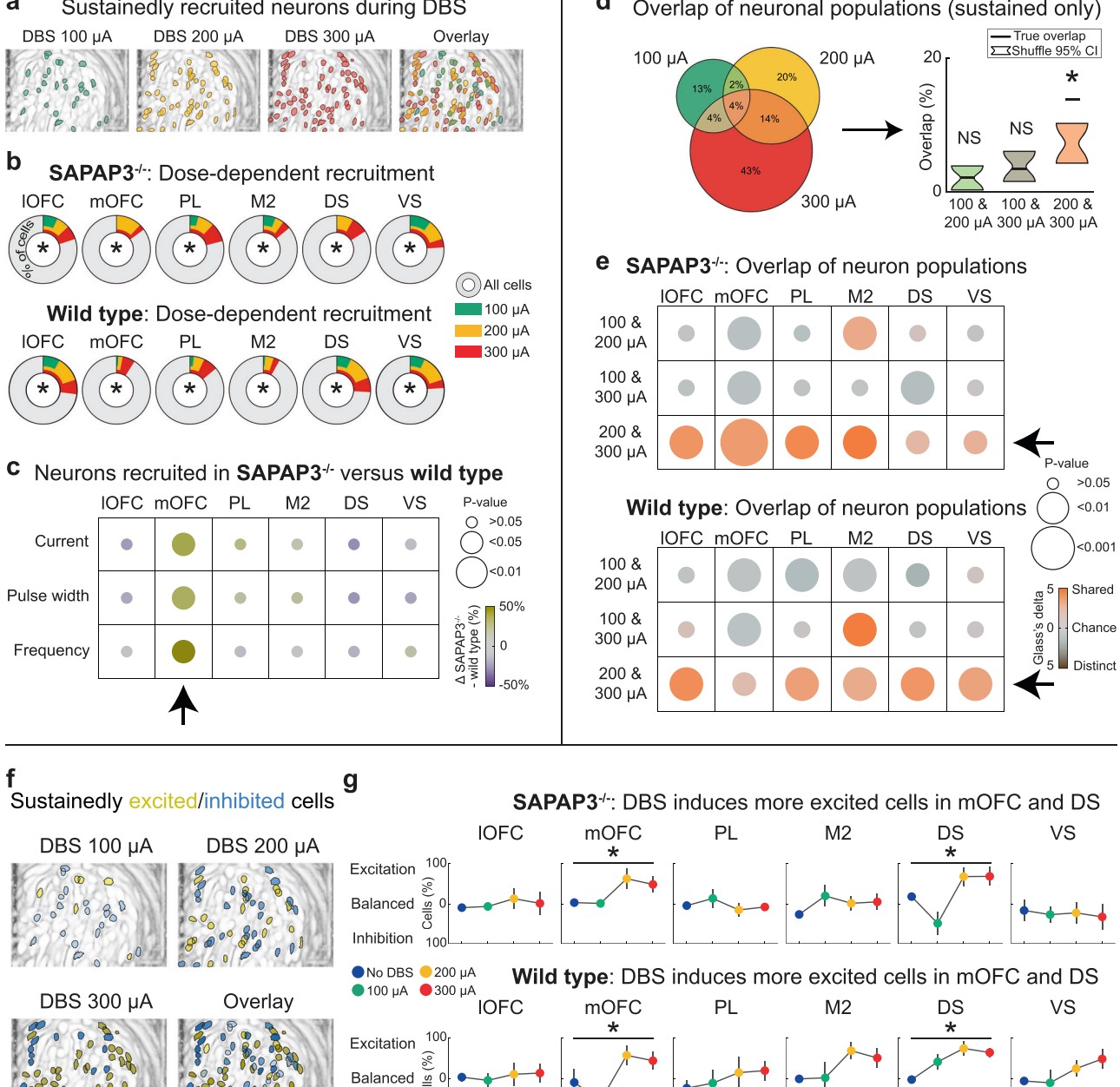

**Fig. 5 | IC-DBS dose-dependently recruits partially overlapping sustained neuron populations, while maintaining excitation/inhibition balance.**
**a** Anatomical distribution of imaged neurons revealed by maximum-intensity projection of neurons recruited during 100 μA, 200 μA, 300 μA, and overlay thereof (example animal). **b** In all recorded regions, we found dose-dependent recruitment of sustained neurons in SAPAP3[-/-] and WT. **c** DBS recruited significantly more neurons in mOFC of SAPAP3[-/-] compared to WT across all DBS-parameter experiments (arrow). **d** Venn-diagram depicts overlap of neuron populations recruited across different current intensities (example region) (left). True percentage of overlapping neuron populations (black horizontal lines) for the different current blocks compared to chance (95% confidence intervals, bootstrap) indicated recruitment of similar neuron populations during the 200 and 300 μA blocks.

**e** For both SAPAP3[-/-] and WT, all cortical and striatal regions showed significant overlap in DBS-recruited neuron populations for the 200 and 300 μA blocks (arrows). **f** Maximum-intensity projection of sustained neurons recruited as excited (yellow) or inhibited (blue) during 100 μA, 200 μA, 300 μA, and overlay thereof (example animal). **g** For both SAPAP3[-/-] and WT, we found an increased number of excited neurons in mOFC and DS during DBS. In all other regions, the number of excited and inhibited neurons were balanced during DBS (SAPAP3[-/-]: lOFC ($n = 6$), mOFC ($n = 5$), PL ($n = 5$), M2 ($n = 5$), DS ($n = 4$), VS ($n = 5$); WT: lOFC ($n = 5$), mOFC ($n = 5$), PL ($n = 5$), M2 ($n = 4$), DS ($n = 5$), VS ($n = 4$). Data are mean ± SEM. Source data are provided as a Source Data file. CI=confidence interval, *$p < 0.05$, NS not significant.

individual neurons (defined as significant activity change) were stable across varying DBS parameters and 2) were dependent on animals being awake. 3) Recruited neurons were scattered throughout brain regions, and 4) their numbers increased with DBS intensity. 5) DBS recruited overlapping yet distinct neuron populations at therapeutic DBS intensities, and 6) the relative balance between the number of neurons excited or inhibited by DBS (E/I balance) was generally preserved, as were baseline activity of individual neurons and their spatiotemporal correlations of activity. The "regional" effects of DBS, those that varied depending on genotype and brain region, were: 1)

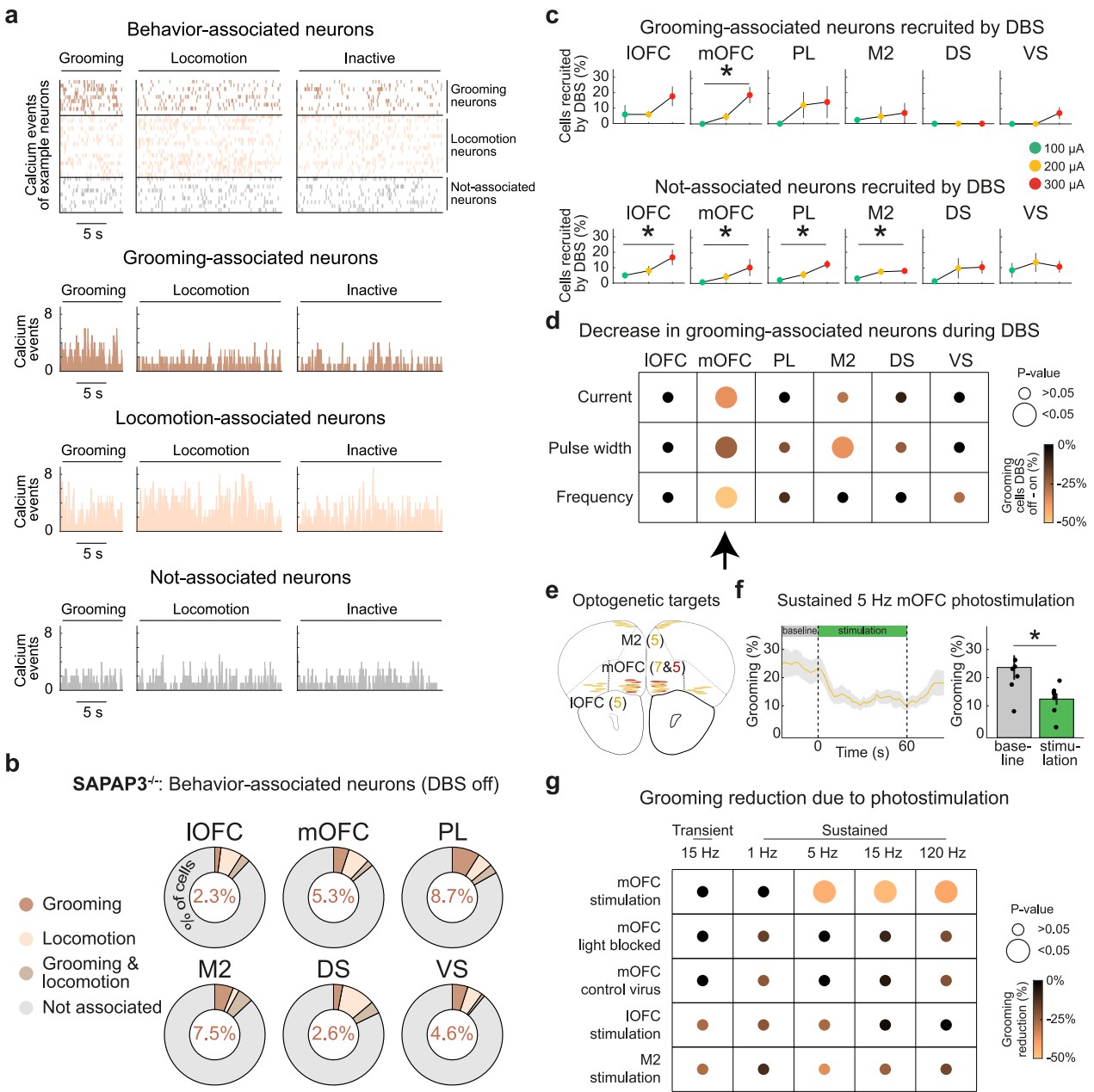

**Fig. 6 | DBS reduced grooming-associated neurons in the mOFC of SAPAP3⁻/⁻, as validated with optogenetics. a** Behavior-associated neurons classified as grooming, locomotion, grooming and locomotion, or not-associated neurons using Bayesian ANOVAs (example animal displayed). Raster plot of the deconvolved calcium events of all behavior-associated neurons across time (top). Histograms show the number of binned calcium events for grooming-associated (top, brown), locomotion-associated (middle, pink), and not-associated (bottom, gray) neurons per behavioral period (i.e., grooming, locomotion, or inactive). **b** In each recorded cortical and striatal region, we detected grooming-, locomotion-, and grooming&locomotion-associated neurons (percentage of grooming-associated neurons depicted in donut charts). **c** DBS recruited grooming-associated neurons in a dose-dependent manner in mOFC (top). DBS did recruit not-associated neurons dose-dependently in all recorded cortical regions (bottom) (SAPAP3⁻/⁻: lOFC ($n = 6$), mOFC ($n = 5$), PL ($n = 5$), M2 ($n = 5$), DS ($n = 4$), VS ($n = 5$); WT: lOFC ($n = 5$), mOFC

($n = 5$), PL ($n = 5$), M2 ($n = 4$), DS ($n = 5$), VS ($n = 4$). **d** DBS reduced the number of grooming-associated neurons in mOFC consistently across DBS-parameter experiments (arrow). **e** Schematic of bilateral optogenetic stimulation of the mOFC ($n = 7$), lOFC ($n = 5$), and M2 ($n = 5$) with ChETA (yellow elliptic shapes), and mOFC ($n = 5$) with mCherry (red elliptic shapes) in SAPAP3⁻/⁻, projected on the Allen Reference Atlas - Mouse Brain[101]. Horizontal elliptic shapes represent the tip of the implanted optical fibers. **f** Reduced grooming during 5 Hz optogenetic stimulation of ChETA-expressing neurons in the mOFC ($n = 7$) (left). Quantification of reduced grooming (dots are individual animals) (right). **g** mOFC photostimulation-induced reduction of grooming using 5, 15, and 120 Hz stimulation frequencies, where blocking the optic fibers abolished the effects on grooming. No effect of photostimulation on grooming was found in control animals (injected with stable fluorophore in mOFC), or after activating lOFC or M2. Data are mean ± SEM.

DBS modulated more mOFC neurons in SAPAP3⁻/⁻ (compared to WT), 2) the E/I balance in mOFC leaned toward excitation, and 3) DBS reduced the number of grooming-associated neurons specifically in mOFC of SAPAP3⁻/⁻. Optogenetically mimicking such regional DBS effects in the mOFC, but not in lOFC nor M2, was sufficient to reduce

excessive grooming in SAPAP3⁻/⁻. Together, our findings provide mechanistic insight into how DBS exerts its anti-compulsive effects.

It has been suggested that IC-DBS exercises its efficacy via modulation of cortico-striato-thalamo-cortical circuits, predominantly via recruitment of cortical regions[2,4,25–29]. Consistent with studies in

patients[25,28,30,32,34,35], we report IC-DBS-driven neuronal recruitment in cortical and striatal regions in mice. Both the number of neurons recruited in these regions and the accompanying reduction in excessive grooming in SAPAP3[−/−] were DBS-dose dependent, suggesting that the extent of neuronal recruitment is related to the behavioral effects. However, such dose-dependent DBS effects occurred exclusively in response to alterations in current and pulse width, but not frequency, validating clinical practice of varying current and pulse width in search of optimal DBS parameters[10]. Furthermore, clinical procedures commonly employ continuous DBS to treat patients, but a recent study investigating DBS in Parkinsonian mice reported improved efficacy of cyclic stimulation (oscillating periods of DBS on and off)[60]. In contrast, we found no effect of such cyclic stimulation on compulsive-like behavior in SAPAP3[−/−], which is consistent with a lack of therapeutic effect in a recent study in OCD patients[48]. In addition, we examined different low-frequency stimulation settings, which also lacked neuronal and behavioral effects. Together, our results indicate that IC-DBS applied both continuously and with high frequency specifically mitigates compulsive-like behavior, where enhanced efficacy can be achieved by increasing current and pulse width, but not frequency.

A striking finding was that although DBS exerted strong effects on brain and behavior, it did not alter basic brain function. For example, baseline neuronal activity (akin to resting-state activity) was not altered[55], and the relationship between anatomical distance and activity of pairs of neurons (i.e., spatiotemporal correlations) was preserved in all recorded regions[56,57]. Furthermore, DBS-induced excitation and inhibition was remarkably balanced in most regions (i.e., equal numbers of excited and inhibited neurons), indicating that the E/I ratio of cortico-striatal networks was relatively unaffected by DBS. Together, this suggests that DBS recruits sustained activity of individual neurons by modulating brain-wide networks without interfering with basic network firing statistics. In support of this conclusion, sustained neuronal recruitment was absent in the anesthetized state (which is characterized by a lack of network activity[54]), illustrating that DBS-induced recruitment of sustained neurons was not simply a product of antidromic entrainment of IC white-matter, but instead depended substantially on network activity in cortico-striato-thalamo-cortical circuits of awake animals. The small percentage of neurons that were nonetheless recruited in anesthetized mice (transiently excited neurons) did overlap with recruited neurons in awake mice, suggesting that such transiently excited neurons may have been recruited by direct, antidromic stimulation of IC white-matter. Moreover, to our surprise, although significantly above chance, only partially overlapping neuron populations were recruited at therapeutic intensities (i.e., consistently across brain regions). We interpret this as another indication of DBS network effects because the electrode position presumably remained stable relative to the surrounding IC axons, and thus variance in identity of recruited cells is at least partially determined by other factors than entrainment or direct disturbance of IC axonal activity. However, in contrast, the direction (excitation or inhibition) and duration (transient or sustained throughout DBS epoch) of modulation of recruited neurons were consistent across different DBS intensities. Together, this suggests that although which neurons are recruited appears predominantly network-dependent, neurons appear to have a predisposition to being recruited in a particular manner, independent of overall network activity. In summary, our findings imply that DBS does not (sustainedly) recruit individual neurons in cortico-striatal regions exclusively via direct antidromic stimulation, but recruits them to a substantial degree indirectly via modulation of broad network activity without compromising basic network function.

The historically influential "inhibition hypothesis" posits that DBS has clinical efficacy because it produces a functional lesion by inhibiting neuronal activity in the vicinity of the electrode[12]. However, electrophysiological studies demonstrate that DBS can also cause excitation of neurons locally[16–18]. Thus, more recently, it has been hypothesized that DBS may generate an "information lesion": Instead of exclusively inhibiting neuronal tissue locally (inhibition hypothesis), DBS may diminish the variability of neuronal-spiking output overall[14], which may promote both excitation and inhibition in the vicinity of the electrode[16–18]. Consistently, a recent study reported DBS-evoked membrane depolarization in neurons proximal to the electrode tip on one hand and interference with somatic action potentials on the other hand[15]. We also detected both DBS-evoked excitation and inhibition, albeit recorded in brain regions distal to the DBS electrodes. Nonetheless, since the IC carries both cortical afferents and efferents (assuming local effects of monosynaptic (mainly excitatory) or antidromic nature, respectively)[3,61,62], the inhibition hypothesis would have predicted an increasingly unbalanced E/I ratio with increasing DBS intensities, which we did not observe. Similarly, it would have predicted disturbed neuronal baseline activity and disrupted spatiotemporal correlations of activity. Furthermore, we report little overlap in recruited neuron populations between different DBS intensities. If DBS were to exclusively inhibit local elements (more and more across increasing DBS intensities), it would be conceivable to expect greater overlap in cell-identity in distal regions than we observed. Finally, we demonstrate that DBS obstructs the encoding of grooming in mOFC neurons, which may be a consequence of DBS interfering with neurons' capacity to accurately respond to synaptic input, a feature of the "information lesion" theory. Collectively, our findings are in agreement with the "information lesion" theory.

The mOFC has been implicated in compulsive behavior in OCD[63,64], and therapeutic IC-DBS modulates mOFC activity and its projections[35]. Consistently, we identified a number of regional effects of DBS in SAPAP3[−/−] that indicate an exceptional role for the mOFC in anti-compulsive effects of DBS: in comparison to all other recorded regions, 1) mOFC was predominantly recruited in SAPAP3[−/−], 2) recruited mOFC neurons were more likely to be excited by DBS than inhibited, and 3) DBS reduced the number of grooming-associated neurons specifically in mOFC. Our optogenetic experiments demonstrate that excitation of mOFC neurons (mimicking DBS effects), but not lOFC nor M2 neurons, reduced excessive grooming (similarly to DBS). Previous rodent studies have linked subregions of the OFC to compulsive-like behaviors[40,45]. Ahmari et al reported that repeated photostimulation of mOFC terminals in the striatum of WT mice induced excessive grooming[65] and Burguière et al demonstrated that photostimulation of the lOFC and its striatal terminals reduced excessive grooming in SAPAP3[−/−][37]. However, in contrast to our work, these studies applied optogenetic stimulation for extended durations (five and three minutes, respectively) and the targeted OFC subdomains differed from ours. These subdomains are functionally heterogeneous[66], which may explain the modest discrepancy in results. In addition, a recent study reports ketamine-induced changes in dorsomedial PFC projections to the striatum in SAPAP3[−/−], as well as reduced grooming in response to direct photostimulation of this pathway[41]. This suggests that different therapeutic interventions and brain targets may normalize behavior via distinct PFC sub-circuits, where our data outlines a potentially important role for mOFC neurons in the therapeutic effects of DBS in compulsive-like behavior.

Although there are obvious methodological advantages to studying DBS mechanisms of action in rodents, it is important to recognize potential limitations. SAPAP3[−/−] resemble core OCD features such as compulsive-like grooming[37–41], increased anxiety[38,42], cognitive impairment[40,42–44], and cortico-striatal deficits[38,67], and respond well to first-line OCD pharmacotherapy[38,45] and DBS[46]. However, the resemblance to OCD is restricted to phenotype, as genome-wide association studies have not flagged the SAPAP3 gene in OCD patients[68,69]. In addition, whether excessive grooming in SAPAP3[−/−] is truly compulsive (i.e., including an urge to perform the behavior), has recently been questioned by a study suggesting that the excessive grooming

contains both compulsive-like and tick-like elements - although it remains unclear how these elements relate to each other[70]. Another potential discrepancy to OCD patients is that we detected immediate grooming reduction upon DBS onset, while OCD patients initially show long periods of DBS-parameter and precise DBS-target optimization[10]. However, following successful optimization, reinstatement of paused DBS immediately alleviates recurrent compulsivity in patients[71], suggesting that DBS's efficacy is dependent on targeting specific IC fibers[72,73]. Thus, this species discrepancy could potentially be explained by a relatively larger sphere of DBS current distribution in mice that captures all critical fibers immediately[3,61,62]. Finally, there are neuroanatomical differences in PFC structure between rodents and primates, where the primate PFC is more extensive[74]. Thus, findings in rodent models cannot speak to the importance of PFC regions that lack a rodent homologue[74,75]. However, directly modulating the IC in both OCD patients and in SAPAP3[−/−] is effective in reducing symptoms[9,46], suggesting that human compulsivity and rodent compulsive-like behavior may share a sufficiently similar neuroanatomy.

Our findings provide novel insights into the mechanisms of action of DBS. We identified a large number of IC-DBS effects which may be relevant for OCD as well as other psychiatric disorders treated with IC-DBS (e.g., addiction, anorexia nervosa, and mood disorders)[2]. We speculate that the therapeutic efficacy of IC-DBS in multiple psychiatric disorders is associated with modulation of dysfunctional activity by exerting widespread (global) effects recruiting neurons scattered throughout multiple frontal cortico-striatal circuits, without over-exciting neural tissue due to the maintained E/I balance. This rather non-specific recruitment of neurons and network nodes may reduce compulsivity as well as potentially alleviate symptoms associated with other psychiatric disorders. Thus, our findings pave the way for further investigations into which effects are associated with what type of therapeutic utility. Our mOFC findings may inspire further clinical exploration of mOFC activity as a readout for DBS-parameter optimization in OCD and other compulsivity disorders, and after further investigation may eventually be used as a biomarker for closed-loop DBS. If successful, such a biomarker has the potential to significantly shorten DBS-parameter optimization periods, improve DBS efficacy overall, and be used to explore novel DBS target regions.

## Methods

### Experimental animals
Male and female SAPAP3 mutant mice (SAPAP3[−/−], $n = 30$) and their wild-type littermates (WT, $n = 28$) were used for deep-brain stimulation (DBS) and imaging experiments, and Thy1−5.17 GCaMP6f mice ($n = 5$)[49] for wide-field experiments. Surgery-naive animals (SAPAP3[−/−], $n = 9$) were used to control for potential effects of GRIN lens-implantation on grooming. For optogenetics experiments, male and female SAPAP3[−/−] ($n = 22$) were used. Animals were housed in a temperature- (21–23 °C) and humidity-controlled (40–65%) room under a 12-hour reversed light/dark cycle with ad-libitum access to food and water (20–50 g; 2-8 months old with an average of 4 months). After surgery, mice were housed solitarily and bedding material for nest building was provided. All experiments were in accordance with Dutch and European laws and approved by the Animal Experimentation Committee of the Royal Netherlands Academy of Arts and Sciences.

### Miniscope imaging and DBS surgery
For calcium imaging experiments employing miniaturized fluorescent microscopes (so-called miniscopes[53]), animals were anesthetized with isoflurane (3%), placed on an isothermal pad to maintain body temperature (37 °C), and placed into a stereotactic frame (Kopf Instruments, USA). Anesthesia was maintained at 1.5% isoflurane (flow rate: 0.6 ml/min O$_2$/air mixture). The head was shaved and disinfected using 70% ethanol. The analgesic drug Metacam (10 mg/kg), non-steroidal anti-inflammatory drug (NSAID) dexamethasone (diluted

2 mg/kg), and saline to prevent dehydration (100 ml/kg), were injected subcutaneously. An incision was made in the skin and lidocaine (100 mg/ml, Astra Zeneca, UK) was applied to the exposed skull and the periosteum removed. Skull was leveled in anteroposterior (AP) and mediolateral (ML) direction before marking the coordinates for unilateral GRIN-lens (left or right hemispheres were counterbalanced across animals) and bilateral internal-capsule (IC) DBS-electrode placements. GRIN lenses were placed in lateral (AP: 2.8 mm, ML: ±1.5 mm, DV: −2.2 mm) or medial (AP: 2.6 mm, ML: ±0.5 mm, DV: −2.2 mm) orbitofrontal cortex (lOFC: SAPAP3[−/−] $n = 6$, WT $n = 5$; mOFC: SAPAP3[−/−] $n = 5$, WT $n = 5$), prelimbic cortex (PL: SAPAP3[−/−] $n = 5$, WT $n = 5$) (AP: 2.1 mm, ML: ±0.3 mm, DV: −1.9 mm), premotor cortex (M2: SAPAP3[−/−] $n = 5$, WT $n = 4$) (AP: 2.3 mm, ML: ±0.35 mm, DV: −0.3 mm), dorsal (AP: 1.1 mm, ML: ±1.5 mm, DV: −2.7 mm), or ventral (AP: 1.1 mm, ML: ±1.1 mm, DV: −4.8 mm) striatum (DS: SAPAP3[−/−] $n = 4$, WT $n = 5$, VS: SAPAP3[−/−] $n = 5$, WT $n = 4$)[76]. After drilling holes, the skull was cleaned and dried, and covered with a layer of bone-attaching cement (SuperBond C&B, Sun Medical Co., LTD, Japan). To improve imaging quality[77,78], we first slowly lowered a 25 G needle (300 nm/min) using a custom-made stereotactic motorized arm (https://osf.io/w7qte/), left it positioned at the target location for 5 min, and then slowly retracted it (300 nm/min). Next, we injected 500 nL virus (200 nL/min, two injections of 250 nL) with a stereotact-mounted syringe (Hamilton, USA) ~100 μm off imaging-target center and waited 5 min per injection to maximize diffusion of virus before retracting the syringe. For cortical-imaging experiments, we injected AAV-DJ-CaMKIIa.GCaMP6s (titre: $3 \times 10^{12}$ vg/ml, diluted 1:10, Stanford University Gene Vector and Virus Core) to express the calcium indicator GCaMP6s in putative pyramidal neurons[79]. For striatal-imaging experiments, we used AAV-DJ-hSyn-GCaMP6s (titre: $5 \times 10^{12}$ vg/ml, diluted 1:5, Stanford University Gene Vector and Virus Core) to predominantly target inhibitory neurons (of which 95% are medium spiny neurons)[79,80]. Using our motorized arm, we lowered (100 nm/min) the GRIN relay-lens (0.6 mm diameter, ~7.3 mm long, Inscopix, USA), covered the gap between lens and skull with cyanoacrylate glue (Bison, The Netherlands), and used cranioplastic cement to secure the lens to the skull. Subsequently, we lowered custom-made DBS electrodes bilaterally into the IC (AP: −0.46 mm, ML: ±1.8 mm, DV: −4.6 mm)[46,81,82], cemented DBS connectors to the skull using cranioplastic cement, and cemented a custom-made head bar to the skull. The GRIN relay-lens was covered and protected using Twinsil speed (Picodent GmbH, Germany). For M2 imaging, we additionally gave animals a subcutaneous injection of 15% D-Mannitol in saline (22 ml/kg) to aid diffusion of virus particles and reduce swelling of the brain[83], injected four times 125 nL virus ~100 μm off imaging-target center and directly placed the GRIN objective-lens (1.8 mm diameter, Edmund Optics Ltd., UK) onto the brain. After surgery, animals received carprofen-analgesic containing drinking water (0.06 mg/ml) for three consecutive days. Animals were allowed to recover for one week.

### Wide-field imaging and DBS surgery
We used Thy1−5.17 GCaMP6f mice that express GCaMP6f throughout the cortex to image the entire dorsal cortex of one hemisphere[84,85]. Animals underwent similar surgery steps as described above ("Miniscope imaging and DBS surgery"). In addition, we applied a thin layer of cyanoacrylate glue (Bison, The Netherlands) to the skull, making the bone transparent[50]. One hole was drilled contralateral to the to be imaged hemisphere (AP: −0.7 mm, ML: −1.72 mm, DV: −5.48 mm) and the DBS electrode was inserted in a 40° angle to target the contralateral IC. For stability and to reduce light glare, we applied a layer of clear cement (SuperBond C&B, Sun Medical Co., LTD, Japan), followed by nail polish (Electron Microscopy Sciences, England). A head bar was placed posterior to lambda, and the outer edges of the clear skull were covered with a small wall of cement (Charisma, Kulzer, Germany) to prevent skin growth. After surgery, animals received

carprofen-analgesic containing drinking water (0.06 mg/ml) for three consecutive days. Animals were allowed to recover for one week.

## Optogenetics surgery

For optogenetics experiments, SAPAP3$^{-/-}$ underwent similar surgery steps as described above ("Miniscope imaging and DBS surgery"). In addition, we bilaterally injected 500 nl virus (200 nL/min, two injections of 250 nl per hemisphere) with a stereotact-mounted syringe (Hamilton, USA) ~100 μm off fiber-target center and waited 5 min per injection to increase diffusion of virus before retracting the syringe. We injected AAV-DJ-hEF1a.ChETA.eYFP (titre: $1.6 \times 10^{12}$ vg/ml, Stanford University Gene Vector and Virus Core) in the mOFC ($n = 7$; AP: 2.6 mm, ML: ±0.5 mm, DV: −2.2 mm), lOFC ($n = 5$; AP: 2.8 mm, ML: ±1.5 mm, DV: −2.2 mm), or M2 ($n = 5$; AP: 2.3 mm, ML: ±0.7 mm, DV: −0.3 mm). To control for nonspecific light effects, we injected AAV8.CaMKIIα.m-Cherry (titre: $1 \times 10^{12}$ vg/ml, Zurich Viral Vector Facility) in the mOFC ($n = 5$). We targeted custom-made optic fibers (FP200URT: 200 μm diameter, 0.5 NA, Thorlabs GmbH, Germany) 200 μm above the injection site (in a 10° angle for mOFC and M2), covered the gap between the fibers and skull with cyanoacrylate glue (Bison, The Netherlands), used cranioplastic cement to secure the fibers to the skull, and made the headcap light proof by painting the outside with black nail polish. After surgery, animals received carprofen-analgesic containing drinking water (0.06 mg/ml) for three consecutive days. Animals were allowed to recover for one week.

## Baseplating

Miniscopes were prepared for deep-brain imaging by drilling a hole in the housing, inserting a screw, and mounting a GRIN objective-lens with a custom-made 3D-printed spacer. Three weeks after surgery, animals were habituated to the experimenter for five consecutive days, followed by three days of habituation to the custom-made head-fixation device with running belt. Animals were head fixed to improve imaging quality during baseplating. During baseplating, the Twinsil-speed protective layer was removed and GRIN lens cleaned using lens paper. A baseplate was mounted onto the miniscope, which was mounted onto a stereotactic arm to hover over the implanted GRIN lens in order to find the best field-of-view. Once an optimal field-of-view was established (i.e., maximizing the number of visible neurons), the baseplate was cemented to the headcap and made light proof by painting the outside with black nail polish. The miniscope was removed and a protective cap installed on the baseplate to avoid damage to the GRIN lens.

## DBS application

DBS electrodes were custom-made and consisted of two bipolar twisted teflon-coated platinum/iridium wires (diameter: 112 μm; bare-wire: 75 μm; distance between the two poles: 0.5 mm). Mice were tethered to deliver DBS via a rotary joint (Adafruit, USA), allowing free unrestricted movement of the animals. DBS parameters were programmed in a digital stimulator (DS8000, WPI, USA) and generated by isolators (DLS100, WPI, USA). DBS settings were inspired by clinical parameters used in OCD patients at the Amsterdam University Medical Centers (Amsterdam UMC, location Amsterdam Medical Center, The Netherlands) and by our previous work[10,46]. DBS pulses were always biphasic, and depending on the experiment, one of the following three parameters was varied systematically while the other two were held constant: current (100, 200, or 300 μA), pulse width (40, 80, or 160 μs (total duration of both phases of the biphasic pulse)), or frequency (60, 120, 180 Hz). The standard DBS parameters were (two of which were always held constant): 200 μA current, 80 μs pulse width, and 120 Hz frequency. All mice underwent these three experiments, which occurred on different days (with weeks in between). In addition, we tried novel stimulation parameters: low frequency (1, 5, 20 Hz) and cyclic (DBS ON (200 μA, 80 μs, 120 Hz) for 10 s, OFF for 1, 5, or 10 s).

## Experimental setup for wide-field imaging experiments

Three weeks after wide-field imaging surgery, animals were habituated to the experimenter for five consecutive days, followed by three days of habituation to the custom-made head-fixation device with running belt. During imaging, mice were head fixed on a stable platform and placed under a wide-field fluorescence microscope (Axio Zoom.V15, ZEISS, Germany) to image the entire dorsal cortex of one hemisphere. Images were captured at 20 Hz (50 ms exposure), stored in 12-bit, $1600 \times 1600$ pixel images (~15 μm per pixel), imaged by a high-speed sCMOS camera (pco.edge 5.5, PCO, Germany), and recorded using Encephalos software (Caenotec). Using an Arduino, the imaging computer triggered the digital stimulator to start and stop DBS.

## Experimental setup for DBS, miniscope imaging, and optogenetics experiments

Experiments were performed in two open fields (custom-made square, light-shielded Perspex boxes, $30 \times 30 \times 40$ cm) housed inside sound-attenuated chambers. Videos were recorded with a Basler GigE camera (monochrome 1/2" Basler acA1300-60gm) attached to a Kowa lens (1/1.8", F 1.6, 4.4–11 mm) and an IR-pass filter (43 mm, $P = 0.75$ mm), mounted in the center above (50 cm) the open field. Two infrared beams illuminated the open field from above and two infrared beams were mounted below the open field (IR-56, Microlight) to illuminate the open field through the transparent floor from below, creating strong contrast between animal and background. Behavioral videos were captured at 30 frames per s, with $1024 \times 768$ pixels, and stored in uncompressed AVI format using a custom-written script in the open-source software Bonsai[86]. A central computer controlled the cameras in both open fields (Dell T3500 workstation, Windows 7 64-bit), while also triggering the digital stimulator to start and stop DBS (via an Arduino), and triggered miniscope data acquisition cards to start and stop calcium imaging (via an Arduino), or triggered blue lasers (DPSS 473 nm, Shanghai Laser & Optics Century Co., Ltd., China) to start and stop photostimulation (via an Arduino). The central computer recorded the (behavioral) video frames and corresponding time stamps, sent TTL triggers (to trigger DBS, imaging, and photostimulation), and saved corresponding TTL trigger time stamps. Using these time stamps, we were able to align DBS, miniscope imaging, and photostimulation data to behavioral data.

## Miniscope-imaging sessions

Mice were habituated to the open fields (custom-made square, light-shielded Perspex boxes, $30 \times 30 \times 40$ cm) for three sessions by placing them in the center of the open field and allowing them to move around freely for 30 min. Animals were head fixed briefly on the running belt for cleaning of the GRIN lens, attaching the miniscope to the baseplate, and connecting the animal to the DBS stimulator. We employed a 6-channel rotary joint (Adafruit, USA) to employ miniscope imaging and DBS in freely moving mice, which was held by a custom-made balancing arm to relieve weight of the animal's head. To minimize bleaching of the calcium sensor in neurons, we imaged animals once a week, maxed out the miniscope sensor's gain, and provided as little excitation LED as possible (0.5–10%). Each session consisted of four DBS blocks (e.g., during the current experiment: 0, 100, 200, or 300 μA stimulation conditions), with eight trials per block. Each trial consisted of 80 s of calcium imaging and 60 s of DBS, starting after 10 s of calcium imaging and ending 10 s before the end of calcium imaging. A fixed inter-trial-interval of 10 s was used between trials within a block, and an interval of 30 s between blocks. Systematic manipulation of a given DBS parameter (four blocks) was tested in a single recording session and animals were exposed to one recording session per week.

The order of stimulation conditions within a session was determined by a Latin square design.

## Optogenetics sessions

Three weeks after surgery, animals were habituated to the experimenter for five consecutive days, followed by three days of habituation to the open fields (see above). Animals were tethered to a blue laser (DPSS 473 nm, Shanghai Laser & Optics Century Co., Ltd., China) via an optical rotary joint (1 × 2 fiber-optic rotary joint, Doric, Canada), which was held by a custom-made balancing arm to relieve weight of the animal's head. Comparable to the DBS experiments, each session consisted of five optogenetics stimulation blocks (15 Hz for 5 s "transient" stimulation or 1, 5, 15, or 120 Hz for 60 s "sustained" stimulation), randomized across animals, with eight trials per condition. Each trial consisted of 120 s of (behavioral) video recording, starting 30 s before optical stimulation (which continued for 60 s in case of sustained stimulation and 5 s in case of transient stimulation) and lasted 90 s after the start of optical stimulation. We administered 5 mW of 473 nm blue light (10 ms pulse-duration) with different frequencies: either 5-s stimulation with 15 Hz to mimic transient activity, or 60-s stimulation with 1, 5, 15, or 120 Hz (4 ms pulse duration) to mimic sustained activity. To control for potential nonspecific effects of photostimulation on behavior, we 1) used animals injected with virus expressing a stable fluorophore (lacking an opsin) and 2) tested animals with ChETA in the mOFC in a condition where laser-light access into the brain was obstructed at the head cap (by a ferrule filled with black nail polish)[87,88].

## Histology

Mice were deeply anesthetized using a lethal dose of pentobarbital, transcardially perfused with 4% PFA in PBS, and decapitated. Heads were submerged in 4% PFA for at least 24 h to preserve lens or fiber and electrode tracks. Subsequently, brains were removed, placed in 30% sucrose for cryoprotection, rapidly frozen using isopentane, and sliced on a cryostat (40 μm coronal sections, −20 °C). Coronal sections containing lens or fiber locations were stained with DAPI to visualize cell nuclei, mounted on glass slides, and imaged with an Axio Scan.Z1 slide scanner (ZEISS, Germany) to validate target location. Sections containing the IC were stained with cresyl violet and imaged with an Axioskop bright-field microscope (ZEISS, Germany) to validate DBS-electrode tip location. Headcaps (GRIN lens and head bar) were placed in acetone for 24 h, cleaned using acetone, ethanol, and lens paper, and reused.

## Modeled sphere of activation

We modeled the current spread around the tip of the DBS electrodes to validate stimulation of IC using the following formula:

$$I = I0 + K \times r^2 \qquad (1)$$

where I is applied current (100 to 300 μA), I0 is amount of current needed to excite an axon (7–22 μA), given that the electrode touches the axon, K is a constant that describes how quickly the threshold current increases as the electrode is moved away from the axon (1292 μA/mm²), and r² is squared distance between the axon and electrode. I0 and K are based on previous studies[89,90]. Using this formula, we found a sphere of activation with a diameter of 0.54 mm (low-intensity DBS) to 0.95 mm (high-intensity DBS).

## Grooming analyses

Grooming behavior was identified by a grooming classifier, as described previously[39]. We trained a Janelia Automatic Animal Behavior Annotator (JAABA) classifier[91] to detect grooming in animals tethered to miniscopes. In short, animal behavior was video-taped and locomotion extracted using Bonsai[86]. Next, we extracted detailed frame-by-frame position information using the open-source software Mouse

Tracker[92], which was fed as input to the JAABA classifier[91]. A human expert observer trained the JAABA grooming classifier on 39.090 frames (19.133 grooming frames and 19.957 not-grooming frames) from eight short videos of SAPAP3[−/−], which provide sufficient amounts of grooming frames. 1/7 Folding cross-validation showed that the classifier was able to reliably detect grooming with 82.3% sensitivity and with 74% specificity. To improve accuracy, we introduced a minimum bout length of 10 frames and set a higher threshold of 0.5 to detect grooming (to reduce false positives). Together, this resulted in 87.2% sensitivity and 92.1% specificity to detect grooming events (Supplementary Fig. 1a–c). Grooming data were binned into 1 s bins and transformed into percentages. To examine whether DBS and photostimulation affected grooming, we used paired t-tests to compare grooming during DBS with grooming before DBS application. In order to explore the relationship between reduction in grooming and DBS electrode location, we calculated relative change in grooming [(grooming during DBS / grooming before DBS) −100] and correlated that with averaged electrode locations in both hemispheres [AP coordinates * DV coordinates].

## Wide-field calcium-imaging analyses

Images were binned into 800 × 800 pixels and converted to a 16-bit format. Images were spatially downsampled by a factor of 2 and registered to the first frame of that session or to the previous session. Subsequently, data were motion corrected within a session by first computing the 2D cross-correlation between the first frame and the remaining frames. Next, frames were rigidly shifted to achieve maximum correlation and inspected manually. Next, we smoothed the data using a Gaussian filter with a standard deviation of two pixels. Per pixel, we calculated relative DF/F, where DF is the activity at a given time and F is the mean activity per trial at 9 to 8.5 s before DBS application. Frames were aligned to the Allen Mouse Brain Common Coordinate Framework using Bregma, Lambda, and suture lines[51]. Regions were frontal cortex (FC), somatosensory cortex (SS), Visual cortex (VIS), and retrosplenial cortex (RSP). Pixels were averaged within these regions. Data were z-scored per region across the entire recording using the following formula:

$$Z = (x − \mu)/\sigma \qquad (2)$$

where Z is the standard score DF/F, x is the observed value, μ is the mean of the region, and σ is the standard deviation of the region. Sustained DBS-induced suppression was examined by statistically comparing z-scored DF/F signals during DBS (last 30 s of the DBS period) across DBS parameters. In addition, to compare between regions, we applied post-hoc tests corrected for multiple comparisons (Tukey's HSD). Finally, we averaged signals across all regions to explore dose-dependent responses across the entire cortex.

## Miniscope calcium-imaging analyses: preprocessing

Calcium-imaging videos were stored as uncompressed AVI files at a rate of 15 frames per s and binned per 1000 frames, and FIJI[93] was used for raw-data inspection. Per animal, we first concatenated all AVI files of a single session into a TIFF file and used logged timestamps of single frames to calculate missing frames. Concatenated TIFF files were motion corrected using NoRMCorre[94] and neuronal footprints and signals extracted using CNMF-E[95]. Data were spatially downsampled by a factor of 2 and we used the following CNMF-E parameters: gSig = 7, gSiz = 17, merge_thr = [1e−1, 0.85, 0], min_pnr = 7.4 (range 4–30), min_corr = 0.8 (range 0.8–0.95). After footprint and signal extraction, we manually cleaned the data using a custom-written user interface that showed spatial footprints of region-of-interests (ROIs) and temporal traces per ROI, calculated distance and correlation between ROIs, and provided the options to delete or merge ROIs. ROIs with artificially small (∼half the size of average ROI) or large (∼twice the size

of average ROI) spatial footprint were discarded as noise or background signal. ROIs with strong overlap (distance <15 pixels, correlation >0.8) were averaged and merged into a single ROIs. We generally used DF/F (C_raw), except for the isoflurane baseline-activity, and behavior-associated neuron analyses. For these analyses, we deconvolved each neuron's DF/F using OASIS to get denoised traces, which were used to estimate calcium events[96]. Using the recorded time stamps, we aligned imaging data to DBS periods, as well as grooming periods. Data were z-scored per neuron across the entire recording session (see above for formula). All analyses were performed in Matlab (R2016b and R2020b, MathWorks Inc., USA).

### Miniscope calcium-imaging analyses: DBS-associated neurons
Trials were divided into pre-DBS baseline period (10 s) and three DBS periods (early (20 s), middle (20 s), late (20 s)). Neurons were classified as responders (recruited neurons) if signal during DBS significantly differed from the pre-DBS baseline signal (paired $t$-test across trials). For transient neurons, only the early DBS period differed from pre-DBS baseline. For sustained neurons, all three DBS periods differed from pre-DBS baseline.

### Miniscope calcium-imaging analyses: consistency of modulation
Single-cell recruitment by DBS varied in duration (transient or sustained) and direction (excited or inhibited) of activity. To examine whether DBS would recruit neurons likewise across stimulation parameters (e.g., 100, 200, and 300 μA), we calculated the consistency of recruitment: We defined consistency of modulation as "neurons in one cluster" divided by "total neurons in that cluster". "Neurons in one cluster" were all the neurons in a given functional cluster that do not fall into another functional cluster across DBS intensities, and "total neurons in that cluster" were all neurons across DBS intensities that were identified as such (e.g., the number of transient excited neurons not found to be sustained excited, or transient or sustained inhibited across other stimulation parameters divided by the total number of transient excited neurons found during DBS). We compared the consistency of recruitment to chance using bootstrapping: From all recorded neurons, we randomly selected the number of neurons as found in the actual data, calculated consistency of modulation, repeated this 1000 times, and calculated summary statistics to compare to the true data.

### Miniscope calcium-imaging analyses: overlap of neurons
We employed Venn diagrams to express overlap of neuron populations. For two overlapping neuron populations, we employed conditional probability to assess the percentage of neurons recruited in condition B, given that they have been recruited in condition A (e.g., the percentage of neurons recruited by DBS under anesthesia, given they have been recruited by DBS in the awake state). Probability can be expressed using the following formula:

$$P(B|A) = P(B \cap A)/P(A) \qquad (3)$$

where P (B | A) are the neurons recruited in condition B, given that they have been recruited in condition A, P (B ∩ A) is the overlap of condition A and B, and P (A) are all neurons recruited in condition A. The overlap of neuron populations was compared to chance level using bootstrapping: From all DBS-recruited neurons across the DBS intensities, we randomly selected the number of neurons as found in each DBS intensity, calculated the overlap between intensities, repeated this 1000 times, and calculated summary statistics to compare to the true data.

### Miniscope calcium-imaging analyses: regional baseline activity
To test whether DBS induced changes in regional baseline activity, we combined the cumulative density function (CDF) of calcium events

and the frequency of activity into a single cell-activity index. We used the size of the calcium events (as used by CDF) but averaged across neurons and binned into one-minute bins (as used to calculate frequency of activity). Per animal, all single-cell deconvolved calcium events were summed up and averaged across neurons to compute the mean number of events per animal during the "no-DBS" block and the high-intensity DBS block. We did this for all animals to compare activity between the "no-DBS" block and the high-intensity DBS block.

### Miniscope calcium-imaging analyses: clustering of recruited neurons
Miniscope imaging provides the spatial location of recorded neurons. To test whether DBS-recruited neurons would cluster (or maximally dispersed), we measured distance to the closest recruited neuron for each recruited neuron. Distance to the closest recruited neuron is important to avoid averaging out short (potential clusters) and long distances (potentially maximally dispersed), which would make the analysis unable to identify any spatial organization. To draw statistical conclusions about spatial modulation, we compared the true distance to chance level (bootstrap: from all recorded neurons, we randomly selected a subset equal to the number of DBS-recruited neurons, measured distance to the closest neuron, averaged over all neuron pairs, repeated this procedure 1000 times, and computed summary statistics to compare to the true closest distance). To validate this analysis method, we ran the analysis on simulated data and were able to identify different forms of clustering (single, multiple, and small clusters), a ring structure, maximally dispersed recruitment, and random recruitment.

### Miniscope calcium-imaging analyses: behavior-associated neurons
Grooming, locomotion (extracted from Bonsai tracking data), and deconvolved data (calcium events, "S") were binned into 333-ms bins (5 bins) to improve estimates of the neurons' activity[97]. Per session, we forced periods of grooming to have no locomotion (set value to 0 in the locomotion vector). Remaining locomotion values were split by their median: All values below median were considered inactive (stationary) and all values above were considered locomotion (Supplementary Fig. 6b). Per neuron, we ran a Bayesian ANOVA[98] with three categories of behavior: grooming, stationary, or locomotion. If the ANOVA was significant and Bayes factor >3, we ran post-hoc tests, corrected for multiple comparisons (Tukey's HSD), to compare activity during the three different behaviors. Classification of neurons was based on the following post-hoc comparisons:

1. grooming-associated neurons: grooming ≠ stationary; stationary = locomotion;
2. locomotion-associated neurons: locomotion ≠ stationary; stationary = grooming;
3. grooming- and locomotion-associated neurons: grooming & locomotion ≠ stationary; grooming = locomotion;
4. not-associated neurons: not significant Bayesian ANOVA and Bayes factor <1/3.

Although a continuous analysis or analysis employing a general linear model would be informative as well, the above-described approach served our purposes best.

### Miniscope calcium-imaging analyses: summary plots
We use summary plots to summarize main effects across different experiments. These plots present four dimensions: 1) The size of the "bubbles" depicts $p$-value (the bigger the bubble size, the lower the $p$-value), 2) color represents measured effect (e.g., change in cell-activity index), 3) columns generally represent the regions recorded (or optogenetic stimulation frequency), and 4) rows generally represent the different experiments (but other variables are possible, too).

## Statistical analyses

Data are presented as mean ± SEM. We used paired and independent *t*-tests, one- or two-way ANOVAs, and bootstrapping to determine statistical significance. A *p*-value of <0.05 was considered statistically significant. When appropriate, the alpha value was adjusted to correct for multiple comparisons (Holm-Bonferroni)[99]. For bootstrapping, we considered the true mean to be significantly different from a bootstrapped chance distribution if the 95th percentile ranges of the two distributions did not overlap. We computed *p*-values using:

$$(1 + X) \div (N + 1) \tag{4}$$

where $X$ represents the number of overlapping data points between two distributions and $N$ number of bootstraps[100]. We calculated effect size using Glass's Delta:

$$X1 - X2 \div SD2 \tag{5}$$

where $X1$ represents the population true mean, $X2$ bootstrapped distribution mean, and $SD2$ standard deviation of the bootstrapped distribution. All statistical analyses were performed using Matlab (R2016b and R2020b, MathWorks Inc., USA).

## Reporting summary

Further information on research design is available in the Nature Portfolio Reporting Summary linked to this article.

## Data availability

The data that support the findings reported in this article and the statistical analyses generated from the data are available on Open Science Framework (https://osf.io/w7qte/). Raw data are available from the corresponding author upon reasonable request. Source data are provided with this paper.

## Code availability

The preprocessing scripts to analyze calcium imaging data are available on Open Science Framework (https://osf.io/w7qte/).

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

## Acknowledgements

We thank Dr. Tycho M. Hoogland, Andres de Groot, Mike Vink, and Joop Bos for technical support for imaging experiments. We are grateful to Ralph Hamelink, Dr. Nicole Yee, and Dr. Arthur S.C. França for technical assistance. We are thankful to Bart Kok and Makaela Weeda for assisting with histology. We thank Dr. Marcus H.C. Howlett, Dr. Matthijs G.P. Feenstra, and Dr. Ester Visser for their comments on the manuscript. We are grateful to the UCLA Miniscope team for sharing the blueprints of the miniscopes. We thank the team of Dr. Karl Deisseroth for advice on the use of viral vectors. We thank Lucia Economico for the mouse illustration. We acknowledge support from the Gravitation program of the Dutch Research Council grant, BRAINSCAPES (024.004.012) (I.W.), and a research grant from FFOR, the Foundation for OCD Research (I.W.).

## Author contributions

B.J.G.v.d.B designed the study, performed the experiments, curated, analyzed, and interpreted the data, wrote the manuscript, and finalized the paper. A.E. and P.A.R. assisted with experiments. E.H.v.B assisted with wide-field imaging experiments and edited the manuscript. A.P. assisted with analyses. D.D. edited the manuscript. I.W. designed and oversaw the study, curated and interpreted the data, wrote the manuscript, and finalized the paper. All authors approved the work.

## Competing interests

The authors declare no competing interests.
