## [Peer Review File · Nature Communications]

Unraveling the mechanisms of deep-brain stimulation of the internal capsule in a mouse modelREVIEWER COMMENTS

Reviewer #1 (Remarks to the Author):

In their manuscript entitled "Unraveling the therapeutic mechanism of deep-brain stimulation", van den Bloom et al. report on a study investigating DBS for OCD in a rodent model (SAPAP3).

The manuscript is well written and timely & will help us get further in understanding the mechanism of action of DBS for OCD.

While I want this to be published, there are a few concerns, mainly regarding framing & interpretation of results.

These points may be helpful in case of a revision:

- The title suggests this study i) covers DBS as a whole (not just DBS to the internal capsule in an OCD model), ii) does not hint at the fact that it's a study in an animal model and iii) suggests we had no clue about how DBS worked beforehand. So I would make the title more specific to OCD-DBS in a mouse model.

- Similar points apply to the general framing of the work. While results are super relevant and add pieces to the puzzle, they are written up in a way as if we "finally understood" how DBS worked based on these results. For instance, authors use their results to refute an "inhibition hypothesis" since they show that activity in some neurons goes up and down in others. To my understanding, the current working model is the one of a 'functional or informational lesion' (I had not heard 'inhibition hypothesis' before being in the field for some time but this could be me). Critically, a functional lesion is not an actual lesion, it just proclaims DBS to 'act as one', on an information theoretical level (e.g. overactivating neurons could still produce an informational lesion since the information channel is jammed with nonsensical information. Hence, this concept (which in my view is the predominant concept most experts currently hold) could not be refuted by measuring

whether activity goes up or down.

Similarly, an actual lesion (e.g. destroying tissue) leads to both decreases and increases in the human brain also. Hence, even though the quality of results is outstanding and we learn a lot, conclusions as drawn by the authors are unfortunately not warranted and their interpretation should be discussed more broadly. (Optimally, showing whether actual lesions leading to similar or different effects would be amazing, but this is probably beyond the scope of present work).

- In the same line, authors speak a lot about "recruitment" of cortical neurons by DBS. This is similarly troublesome, since e.g. an overactivation of a neuron could result from indirect network dynamics after applying DBS to a broadly wired structure such as the IC. The same for inhibitions (reductions in firing rate). In fact, some findings speak *against* these effects to be direct/monosynaptic results from DBS: The I/E ratio and the overall functioning of the networks seem to be maintained under DBS. Isn't it hard to believe that each DBS electrode would know how many neurons to excite and how many to inhibit? Rather, one network seems disturbed which leads to alterations of many brain networks ("entire dorsal cortex") due to changes in the overall functional repertoire of the brain, *not* as a direct function of DBS. I/E is maintained and general wiring is also.

- At some points, authors conclude from absence of significance that e.g. there is no correlation (e.g. lines 82 ff). However, this is not a valid way of thinking and these remarks should be removed. It is for instance well established that both frequency and electrode placement are critical for success (the former e.g. very well established in the very first Benabid Lancet studies), the latter common sense. The fact that authors don't find a significant relationship here is fine & should be reported, but we shouldn't conclude that there is no effect.

- Again, all of the above points address framing & interpretation and could be easily addressed by a broader discussion / less emphasis on the idea that a riddle has finally been solved. I want to emphasize again that I am generally very much in favor of publishing this amazing piece. It is refreshing to see animal work studying treatments and their effects that are actually applied in patients. I learned a lot reading the paper and so will the field.

Reviewer #2 (Remarks to the Author):

This paper uses the SAPAP3 knockout mouse (an alleged model for OCD, or at least aspects thereof) as a vehicle for studying potential mechanisms of high-frequency capsular DBS. It directly studies electrical stimulation of a mouse homologue of the ALIC, using optical recording to get past the artifact. Stimulation mimicked key properties of human DBS, namely that only high frequencies and higher intensities (longer PW/higher current) produced effects. At the wide-field level (which in theory might correspond to human LFP/ECOG/EEG), DBS had a primarily suppressive effect, particularly in frontal regions. At the single unit level (which also captured more ventral PFC regions such as OFC that are invisible to widefield), this is a mix of phasic and tonic excitation and inhibition, and that also shows the dose dependence one would hope for. Interestingly, there is a specific bias towards excitation of medial OFC and dorsal striatum, whereas other regions have a more mixed inhibition-excitation balance. (This is less clear for VS which may have a mild excitation bias.) This is mostly identical between KO and WT animals, i.e. it is an effect of DBS, not an interaction between DBS and the corticostriatal dysfunction of KOs. The exception is in mOFC, where they suggest that DBS has a steeper recruitment curve (more total

cells at each intensity) in KOs. This, combined with the excitation bias, leads to an E/I balance disruption in mOFC. They further demonstrate that DBS' effect may be an encoding disruption – DBS does not preferentially modulate units whose firing is correlated with grooming, but instead reduces the number of such units, somewhat consistent with the “information lesion” hypothesis.

There are also some surprising findings. Counter to what might be expected given ALIC DBS' effect on both striatal and cortical LFP oscillations, firing statistics and inter-neuronal correlation did not change despite this sustained excitation. The neurons activated (for sustained firing) are not consistent as the stimulation intensity increases, i.e. 300 uA is not a strict superset of 200 uA activation. In fact, the majority of neurons pushed into sustained activation by 200 uA are *not* sustained-activated by 300 uA (per Figure 5d).

This is a very technically strong study, and in a well written paper that is fairly easy to follow despite the complexity of the subject matter and techniques. Among its many strengths are a clinically relevant question, very difficult technique, consideration of multiple DBS parameters and systematic exploration thereof, strong controls in all experiments, and generally adequate N. They specifically control for DBS parameters, effects in wildtype, effects from other cortices, and some aspects of animal state (asleep vs awake). Overall, the vast majority of the conclusions are well supported by the presented data (see below for minor exceptions that do not detract from the central argument). There are some additional points they need to discuss and a few small additional analyses just to cement the whole thing together, but there are no major technical flaws. The study as a whole has a lot of potential impact – it does tell us a lot about what DBS does to the PFC, and while there is a translational gap (see next paragraph), I can easily see this paper spurring many conversations and follow up experiments.

There is an underlying weakness in the model itself. They are showing rapid onset/offset of grooming changes. While there are indeed acute effects of ALIC DBS, they are not on compulsive behaviors – they tend to relate to mood and overall energy levels, and as described in some of the cited papers, to acute cognitive changes that patients themselves do not detect. The clinical symptoms of OCD (i.e., the YBOCS) changes over months, and there has never been a strong correlation shown between acute mood effects and that long term recovery. Dr. Denys in fact discusses this in van Westen et al 2021, as do other authors. As such, while the effect they are showing is clearly real, it is questionable about whether it models the slower change seen in human DBS. In general, whether it is the SAPAP model or any other knockout/manipulation, we still have a large gap between what we can do in rodents and the behavior in the clinical disease. Mouse grooming is not functionally equivalent even to human compulsive handwashing. Further, there are major structures that appear to be DBS-modulated and involved in response inhibition (e.g., the VLPFC, as mentioned in Widge et al. 2019, Vaghi et al. Biological Psychiatry 2017, Safadi et al. J Neurosci 2018) that do not exist in rodent, and this model cannot speak to whether they are relevant to DBS response.

None of this is a fatal weakness of the paper; the results are still strong and interesting and mechanistically important. But, the points just mentioned, and the broader point about what mouse models do and do not say about DBS, needs to be clearly laid out in the Discussion. This should not just be a sentence, but a substantial and reasonably referenced independent paragraph. (They might also consider discussing Lamothe et al. Translational Psychiatry 2023, which I realize was not published at the time of this manuscript submission, but which specifically speaks to the SAPAP model.)

Major Comments:

1. See above about the need for a clearer limitations Discussion.
2. Closely related, I would request that they soften/moderate the claims such as “our data outlines a specific role for mOFC neurons in the therapeutic effects of DBS in compulsive behavior”. For instance, perhaps call this a “potentially important role”? Given that we do not quite know that SAPAP grooming is “compulsive” (as opposed to, e.g., more tic-like), it would be more fair to call it “compulsion-like” or “excessive”. We need to avoid anthropomorphizing the animal model.
3. There are a few places where the argument and conclusions will be strengthened by small supplemental analyses that do not require any additional experiments:
 - a. They should demonstrate that there is no grooming behavior difference based on the site of GRIN lens implantation (in WT or KO), given that these do cause very large lesions that may affect behavior. Or, if it is there, they could show that it is much smaller than the effect of DBS.
 - b. They never quite establish that it is the sustained-activated neurons that relate to grooming. That is, there is a bit of an intellectual jump between Figure 3/5 and Figure 6. I would like to see that correlation actually tested, e.g., to see some plot comparing “number of sustained-activated cells for DBS setting X” against “percentage reduction in grooming”. Since they appear to be claiming that this is the mechanism, some correlation needs to be shown, or if it is not there, they need to say why this is reasonable, particularly because they appear to formally make the claim of that correlation at line 299-300.
 - c. They established “locomotion associated neurons” by median splitting the speed. They really should show a distribution to verify that this is reasonable (that there is something representing bimodality and that the median is the correct quantile).
4. They need to discuss more the finding of Figure 5d, that the populations of lower vs higher intensity do not really overlap. Yes, the finding is statistically significant, but a <20% overlap means that most neurons are discordant between current levels. This is surprising, and deserves at least speculation to how it could arise (and needs to be flagged for the generalist reader as very surprising). Does this imply that some neurons are getting “overdosed” and no longer responding? Are they claiming some form of reverberant network effect?

Minor Comments:

5. The title is too strong, especially since at best, this speaks to therapeutic mechanisms of IC DBS for compulsive behavior. “Cortico striatal mechanisms of internal capsule DBS in a rodent compulsivity model” would be a good replacement.
6. As per my comments in summarization above, they might consider citing some of Warren Grill’s work on informational lesions as a DBS mechanism in PD, as possibly the same concept is at play here. Ref 46 is a start but there are some broader theoretical pieces.

7. They need to remove the “trend” claim at line 1051 and 221. There are a number of papers discussing why it is not the case that “with a few more data points this would have been significant”. Having this claim detracts from an otherwise really methodologically and analytically robust paper.

8. It is a little surprising to see in Supp 1 that the behavior effect was not sensitive to electrode placement. One or two sentences of speculation or discussion on why this is would be nice (given the intense focus in the clinical literature on tractographic mapping and targeting). Possibly this is simply a function of how large the probe is relative to the mouse IC, i.e. the volume of activated tissue is always so large as to cover most of the fibers? It sounds like they have actually done this modeling based on the Methods.

9. If they have room for extra references, they could bolster the (correct) point that DBS recruits cortical regions by referencing some of the recent human work combining cortical recording (acute and chronic) with ALIC stimulation, all of which also did show cortical changes and possibly biomarkers of compulsivity. Examples are Olsen et al. *Frontiers* 2019, Figeo et al *Nature Neurosci* 2013 (cited but not in this context), Basu et al *Nature BME* 2021 (not exactly ALIC but close). There are of course also a number of DBS-lead recording papers including their own work.

10. Given that technically in the striatum they recorded inhibitory neurons which are probably but not guaranteed to be MSNs, it would probably be best to describe these as “putative medium spiny neurons”.

11. For the pulse width, could they please specify whether 40/80/160 us is the width of the full pulse (both phases) or solely of the cathodal phase? This causes confusion in the literature, especially since animal work tends to use symmetric biphasic whereas human DBS uses asymmetric pulses and reports only the cathodal width. (Incidentally, line 493 incorrectly gives durations in uA.)

REVIEWER COMMENTS

Reviewer #1 (Remarks to the Author):

In their manuscript entitled "Unraveling the therapeutic mechanism of deep-brain stimulation", van den Bloom et al. report on a study investigating DBS for OCD in a rodent model (SAPAP3). The manuscript is well written and timely & will help us get further in understanding the mechanism of action of DBS for OCD. While I want this to be published, there are a few concerns, mainly regarding framing & interpretation of results.

These points may be helpful in case of a revision:

- The title suggests this study i) covers DBS as a whole (not just DBS to the internal capsule in an OCD model), ii) does not hint at the fact that it's a study in an animal model and iii) suggests we had no clue about how DBS worked beforehand. So I would make the title more specific to OCD-DBS in a mouse model.

We changed the title of the manuscript to describe our work more precisely:

"Unraveling the therapeutic mechanism of deep-brain stimulation of the internal capsule in a mouse model"

- Similar points apply to the general framing of the work. While results are super relevant and add pieces to the puzzle, they are written up in a way as if we "finally understood" how DBS worked based on these results.

For instance, authors use their results to refute an "inhibition hypothesis" since they show that activity in some neurons goes up and down in others. To my understanding, the current working model is the one of a 'functional or informational lesion' (I had not heard 'inhibition hypothesis' before being in the field for some time but this could be me).

Critically, a functional lesion is not an actual lesion, it just proclaims DBS to 'act as one', on an information theoretical level (e.g. overactivating neurons could still produce an informational lesion since the information channel is jammed with nonsensical information. Hence, this concept (which in my view is the predominant concept most experts currently hold) could not be refuted by measuring whether activity goes up or down. Similarly, an actual lesion (e.g. destroying tissue) leads to both decreases and increases in the human brain also. Hence, even though the quality of results is outstanding and we learn a lot, conclusions as drawn by the authors are unfortunately not warranted and their interpretation should be discussed more broadly. (Optimally, showing whether actual lesions leading to

similar or different effects would be amazing, but this is probably beyond the scope of present work).

We agree with the reviewer that the manuscript would benefit from discussing our findings more broadly and toning down the language. We have rephrased our introduction and discussion accordingly.

We now properly introduce and discuss the ‘inhibition hypothesis’, which posits that local inhibition of neuronal activity is the primary DBS mechanism, as mainly historically influential (especially in the field of “motor” disorders), and we cite publications that report electrophysiological recordings that question this hypothesis by demonstrating that DBS can also cause excitation of neurons locally (Agnesi et al., 2013; Chiken and Nambu, 2016, 2014; Dostrovsky and Lozano, 2002; Mohan et al., 2020; Steiner et al., 2021; Udupa and Chen, 2015; Vachez and Creed, 2020; Vitek, 2002).

Furthermore, we expand the discussion of the widely assumed working hypothesis of an ‘information lesion’ (Grill & Snyder, 2004), stating that DBS disrupts neurons’ ability to respond to synaptic input (rather than exclusively suppressing neuronal activity) and thereby creates a disturbance on an “information level” (as remarked by the reviewers).

We believe that there was a misunderstanding, because we did not intend to refute the information-lesion hypothesis in our original submission. We now describe the inhibition and the information-lesion hypotheses more clearly and discuss our findings in relation to these two theories.

Revised introduction:

A historically influential theory postulates that DBS inhibits neural tissue surrounding the electrode tip (“inhibition hypothesis”) and thereby creates a reversible lesion with clinical efficacy similar to irreversible lesions of the subthalamic nucleus in neurological movement disorders and to that of capsulotomy in psychiatry. More recently, a modified version of this idea has been favored: DBS is assumed to act as a (reversible) “information lesion”, potentially by interfering with the ability of neurons to properly respond to synaptic input. Thus, instead of exclusively (and completely) inhibiting neuronal activity, as assumed by the inhibition hypothesis, DBS is thought to create a disturbance on an “information level”, which may promote both excitation and inhibition in the vicinity of the electrode. This is supported by findings that demonstrate that DBS can cause both excitation and inhibition of neurons locally.

Revised discussion:

The historically influential “inhibition hypothesis” posits that DBS has clinical efficacy because it produces a functional lesion by inhibiting neuronal activity in the vicinity of the electrode. However, electrophysiological studies demonstrate that DBS can also cause excitation of neurons locally. Thus, more recently, it has been hypothesized that DBS may generate an “information lesion”: Instead of exclusively inhibiting neuronal tissue locally (inhibition hypothesis), DBS may diminish the variability of neuronal-spiking output overall, which may promote both excitation and inhibition in the vicinity of the electrode. Consistently, a recent study reported DBS-evoked membrane depolarization in neurons proximal to the electrode tip on one hand and interference with somatic action potentials on the other hand. We also detected both DBS-evoked excitation and inhibition, albeit recorded in brain regions distal to the DBS electrodes. Nonetheless, since the IC carries both cortical afferents and efferents (assuming local effects of monosynaptic (mainly excitatory) or antidromic nature, respectively), the inhibition hypothesis would have predicted an increasingly unbalanced E/I ratio with increasing DBS intensities, which we did not observe. Similarly, it would have predicted disturbed neuronal baseline activity and disrupted spatiotemporal correlations of activity. Furthermore, we report little overlap in recruited neuron populations between different DBS intensities. If DBS were to exclusively inhibit local elements (more and more across increasing DBS intensities), it would be conceivable to expect greater overlap in cell-identity in distal regions than we observed. Finally, we demonstrate that DBS obstructs the encoding of grooming in mOFC neurons, which may be a consequence of DBS interfering with neurons’ capacity to accurately respond to synaptic input, a feature of the “information lesion” theory. Collectively, our findings are in agreement with the “information lesion” theory.

- In the same line, authors speak a lot about "recruitment" of cortical neurons by DBS. This is similarly troublesome, since e.g. an overactivation of a neuron could result from indirect network dynamics after applying DBS to a broadly wired structure such as the IC. The same for inhibitions (reductions in firing rate). In fact, some findings speak *against* these effects to be direct/monosynaptic results from DBS: The I/E ratio and the overall functioning of the networks seem to be maintained under DBS. Isn't it hard to believe that each DBS electrode would know how many neurons to excite and how many to inhibit? Rather, one network seems disturbed which leads to alterations of many brain networks ("entire dorsal cortex") due to changes in the overall functional repertoire of the brain, *not* as a direct function of DBS. I/E is maintained and general wiring is also.

We completely agree with the reviewer. Apparently, we were not successful in conveying this opinion in the original submission properly though. Similar to what the reviewer points out, we believe that most of the effects are indeed indirect in nature (i.e., not caused by

direct antidromic stimulation, but rather indirectly via network activity). For example, it is conceivable that the stable E/I ratio is the result of a large network maintaining its status quo (rather than DBS precisely targeting specific neurons). In addition, the spatio-temporal correlations that were unaffected by DBS are likely caused by network inputs to these neurons, rather than an exclusive direct modulation. Most importantly, the lack of sustainedly modulated neurons during the anesthetized state provides additional evidence for DBS exerting its effect via indirect network modulation, rather than directly/monosynaptic modulation of individual neurons. We updated our discussion to more clearly convey this message. Importantly, we now stress our definition of “recruitment” much more (i.e., neurons that increased or decreased their activity compared to baseline in response to DBS onset, independent of whether this activity change was induced directly or indirectly via network effects).

Revised results:

To describe how DBS modulates neuronal activity, we use the term “recruitment”. We define recruitment as a significant change in activity, irrespective of direction of change (increase or decrease), duration of change (transient or sustained), or origin of change (direct entrainment or indirect, network-driven modulation).

Revised discussion:

A striking finding was that although DBS exerted strong effects on brain and behavior, it did not alter basic brain function. For example, baseline neuronal activity (akin to resting-state activity) was not altered, and the relationship between anatomical distance and activity of pairs of neurons (i.e., spatiotemporal correlations) was preserved in all recorded regions. Furthermore, DBS-induced excitation and inhibition was remarkably balanced in most regions (i.e., equal numbers of excited and inhibited neurons), indicating that the E/I ratio of cortico-striatal networks was relatively unaffected by DBS. Together, this suggests that DBS recruits sustained activity of individual neurons by modulating brain-wide networks without interfering with basic network firing statistics. In support of this conclusion, sustained neuronal recruitment was absent in the anesthetized state (which is characterized by a lack of network activity), illustrating that DBS-induced recruitment of sustained neurons was not simply a product of antidromic entrainment of IC white-matter, but instead depended substantially on network activity in cortico-striato-thalamo-cortical circuits of awake animals. The small percentage of neurons that were nonetheless recruited in anesthetized mice (transiently excited neurons) did overlap with recruited neurons in awake mice, suggesting that such transiently excited neurons may have been recruited by direct, antidromic stimulation of IC white-matter. Moreover, to our surprise, although significantly above

chance, only partially overlapping neuron populations were recruited at therapeutic intensities (i.e., consistently across brain regions). We interpret this as another indication of DBS network effects because the electrode position presumably remained stable relative to the surrounding IC axons, and thus variance in identity of recruited cells is at least partially determined by other factors than entrainment or direct disturbance of IC axonal activity. However, in contrast, the direction (excitation or inhibition) and duration (transient or sustained throughout DBS epoch) of modulation of recruited neurons were consistent across different DBS intensities. Together, this suggests that although which neurons are recruited appears predominantly network-dependent, neurons appear to have a predisposition to being recruited in a particular manner, independent of overall network activity. In summary, our findings imply that DBS does not (sustainedly) recruit individual neurons in cortico-striatal regions exclusively via direct antidromic stimulation, but recruits them to a substantial degree indirectly via modulation of broad network activity without compromising basic network function.

- At some points, authors conclude from absence of significance that e.g. there is no correlation (e.g. lines 82 ff). However, this is not a valid way of thinking and these remarks should be removed. It is for instance well established that both frequency and electrode placement are critical for success (the former e.g. very well established in the very first Benabid Lancet studies), the latter common sense. The fact that authors don't find a significant relationship here is fine & should be reported, but we shouldn't conclude that there is no effect.

We thank the reviewer for making this important point and agree that lack of significance is not equivalent to lack of an effect. We agree that electrode placement is obviously an important factor in the therapeutic efficacy of DBS, and this lack of relationship came as a surprise given the focus of clinical literature on tractography mapping and targeting. Therefore, we updated our results and clearly state that the electrodes need to be implanted in the mouse's ventral IC for DBS to be effective. However, we believe that our results indicate that if the ventral IC is properly targeted, the exact location within the ventral IC appears to not exhibit a strong correlation with grooming reduction. To strengthen this interpretation, we ran Bayesian correlations and report substantial evidence against the correlation ($BF > 3$; current: 6.2, pulse width: 3.7, frequency: 4.9). Therefore, we speculate that our DBS parameters induce a large enough sphere of current distribution that influences most ventral-IC fibers irrespective of precise electrode coordinates within the ventral IC. This sphere of current distribution has been modeled and visualized as colored halos in figure 1b.

- Again, all of the above points address framing & interpretation and could be easily addressed by a broader discussion / less emphasis on the idea that a riddle has finally been solved. I want to emphasize again that I am generally very much in favor of publishing this amazing piece. It is refreshing to see animal work studying treatments and their effects that are actually applied in patients. I learned a lot reading the paper and so will the field.

We would like to thank the reviewer for their time and energy in putting together their constructive review. We believe that the manuscript improved substantially due to their comments and hope that their concerns have been addressed sufficiently.

References

- Agnesi, F., Johnson, M.D., Vitek, J.L., 2013. Deep brain stimulation: how does it work? *Brain Stimulat.* 116, 39–54.
- Chiken, S., Nambu, A., 2016. Mechanism of Deep Brain Stimulation: Inhibition, Excitation, or Disruption? *The Neuroscientist* 22, 313–322.
- Chiken, S., Nambu, A., 2014. Disrupting neuronal transmission: mechanism of DBS? *Front. Syst. Neurosci.* 8.
- Dostrovsky, J.O., Lozano, A.M., 2002. Mechanisms of deep brain stimulation. *Mov. Disord.* 17, S63–S68.
- Mohan, U.R., Watrous, A.J., Miller, J.F., Lega, B.C., Sperling, M.R., Worrell, G.A., Gross, R.E., Zaghloul, K.A., Jobst, B.C., Davis, K.A., Sheth, S.A., Stein, J.M., Das, S.R., Gorniak, R., Wanda, P.A., Rizzuto, D.S., Kahana, M.J., Jacobs, J., 2020. The effects of direct brain stimulation in humans depend on frequency, amplitude, and white-matter proximity. *Brain Stimulat.* 13, 1183–1195.
- Steiner, L.A., Kühn, A.A., Geiger, J.R., Alle, H., Popovic, M.R., Kalia, S.K., Hodaie, M., Lozano, A.M., Hutchison, W.D., Milosevic, L., 2021. Persistent synaptic inhibition of the subthalamic nucleus by high frequency stimulation. *BioRxiv*.
- Udupa, K., Chen, R., 2015. The mechanisms of action of deep brain stimulation and ideas for the future development. *Prog. Neurobiol.* 133, 27–49.
- Vachez, Y.M., Creed, M.C., 2020. Deep Brain Stimulation of the Subthalamic Nucleus Modulates Reward-Related Behavior: A Systematic Review. *Front. Hum. Neurosci.* 14.
- Vitek, J.L., 2002. Mechanisms of deep brain stimulation: Excitation or inhibition. *Mov. Disord.* 17, S69–S72.

Reviewer #2 (Remarks to the Author):

This paper uses the SAPAP3 knockout mouse (an alleged model for OCD, or at least aspects thereof) as a vehicle for studying potential mechanisms of high-frequency capsular DBS. It directly studies electrical stimulation of a mouse homologue of the ALIC, using optical recording to get past the artifact. Stimulation mimicked key properties of human DBS, namely that only high frequencies and higher intensities (longer PW/higher current) produced effects. At the wide-field level (which in theory might correspond to human LFP/ECOG/EEG), DBS had a primarily suppressive effect, particularly in frontal regions. At the single unit level (which also captured more ventral PFC regions such as OFC that are invisible to widefield), this is a mix of phasic and tonic excitation and inhibition, and that also shows the dose dependence one would hope for. Interestingly, there is a specific bias towards excitation of medial OFC and dorsal striatum, whereas other regions have a more mixed inhibition-excitation balance. (This is less clear for VS which may have a mild excitation bias.) This is mostly identical between KO and WT animals, i.e. it is an effect of DBS, not an interaction between DBS and the corticostriatal dysfunction of KOs. The exception is in mOFC, where they suggest that DBS has a steeper recruitment curve (more total cells at each intensity) in KOs. This, combined with the excitation bias, leads to an E/I balance disruption in mOFC. They further demonstrate that DBS' effect may be an encoding disruption – DBS does not preferentially modulate units whose firing is correlated with grooming, but instead reduces the number of such units, somewhat consistent with the “information lesion” hypothesis.

There are also some surprising findings. Counter to what might be expected given ALIC DBS' effect on both striatal and cortical LFP oscillations, firing statistics and inter-neuronal correlation did not change despite this sustained excitation. The neurons activated (for sustained firing) are not consistent as the stimulation intensity increases, i.e. 300 uA is not a strict superset of 200 uA activation. In fact, the majority of neurons pushed into sustained activation by 200 uA are *not* sustained-activated by 300 uA (per Figure 5d).

This is a very technically strong study, and in a well written paper that is fairly easy to follow despite the complexity of the subject matter and techniques. Among its many strengths are a clinically relevant question, very difficult technique, consideration of multiple DBS parameters and systematic exploration thereof, strong controls in all experiments, and generally adequate N. They specifically control for DBS parameters, effects in wildtype, effects from other cortices, and some aspects of animal state (asleep vs awake). Overall, the vast majority of the conclusions are well supported by the presented data (see below for minor exceptions that do not detract from the central argument). There are some additional points they need to discuss and a few small additional analyses just to cement the whole

thing together, but there are no major technical flaws. The study as a whole has a lot of potential impact – it does tell us a lot about what DBS does to the PFC, and while there is a translational gap (see next paragraph), I can easily see this paper spurring many conversations and follow up experiments.

There is an underlying weakness in the model itself. They are showing rapid onset/offset of grooming changes. While there are indeed acute effects of ALIC DBS, they are not on compulsive behaviors – they tend to relate to mood and overall energy levels, and as described in some of the cited papers, to acute cognitive changes that patients themselves do not detect. The clinical symptoms of OCD (i.e., the YBOCS) changes over months, and there has never been a strong correlation shown between acute mood effects and that long term recovery. Dr. Denys in fact discusses this in van Westen et al 2021, as do other authors. As such, while the effect they are showing is clearly real, it is questionable about whether it models the slower change seen in human DBS. In general, whether it is the SAPAP model or any other knockout/manipulation, we still have a large gap between what we can do in rodents and the behavior in the clinical disease. Mouse grooming is not functionally equivalent even to human compulsive handwashing. Further, there are major structures that appear to be DBS-modulated and involved in response inhibition (e.g., the VLPFC, as mentioned in Widge et al. 2019, Vaghi et al. Biological Psychiatry 2017, Safadi et al. J Neurosci 2018) that do not exist in rodent, and this model cannot speak to whether they are relevant to DBS response.

None of this is a fatal weakness of the paper; the results are still strong and interesting and mechanistically important. But, the points just mentioned, and the broader point about what mouse models do and do not say about DBS, needs to be clearly laid out in the Discussion. This should not just be a sentence, but a substantial and reasonably referenced independent paragraph. (They might also consider discussing Lamothe et al. Translational Psychiatry 2023, which I realize was not published at the time of this manuscript submission, but which specifically speaks to the SAPAP model.)

Major Comments:

1. See above about the need for a clearer limitations Discussion.

We agree with the reviewer that there are limitations to the use of rodent models. As the reviewer suggested, we added an independent paragraph to the discussion. As recommended, we now also mention Lamothe, et al., 2023 (published after submission of our manuscript) in light of the limitations of SAPAP3 KO mice. Furthermore, we address neuroanatomical differences in PFC structure between humans and rodents, and we discuss the differences in discrepant temporal characteristics of DBS effects on excessive grooming

(mice) and compulsivity (patients). More specifically, while we find immediate effects on grooming in mice, it takes months to optimize DBS parameters and precise targeting to reduce compulsivity in patients. However, once optimal DBS parameters and electrode-contact points are found, reinstatement of DBS immediately alleviates recurrent compulsive symptoms in OCD patients after a DBS pause (de Koning, et al., 2016). Although surprising that DBS in SAPAP3 KO mice does not require such an optimization period to find “optimal DBS parameters”, we speculate that our electrode composition and DBS parameters induce a relatively large sphere of current distribution targeting a relatively large number of IC fibers, resulting in broad network effects that reduce grooming. Figure 1b depicts the modeled sphere of current distribution as colored halos.

Revised discussion:

Although there are obvious methodological advantages to studying DBS mechanisms of action in rodents, it is important to recognize potential limitations. SAPAP3^{-/-} mice resemble core OCD features such as compulsive-like grooming, increased anxiety, cognitive impairment, and cortico-striatal deficits, and respond well to first-line OCD pharmacotherapy and DBS. However, the resemblance to OCD is restricted to phenotype, as genome-wide association studies have not flagged the SAPAP3 gene in OCD patients. In addition, whether excessive grooming in SAPAP3^{-/-} is truly compulsive (i.e., including an urge to perform the behavior), has recently been questioned by a study suggesting that the excessive grooming contains both compulsive-like and tick-like elements - although it remains unclear how these elements relate to each other. Another potential discrepancy to OCD patients is that we detected immediate grooming reduction upon DBS onset, while OCD patients initially show long periods of DBS-parameter and precise DBS-target optimization¹⁰. However, following successful optimization, reinstatement of paused DBS immediately alleviates recurrent compulsivity in patients, suggesting that DBS's efficacy is dependent on targeting specific IC fibers. Thus, this species discrepancy could potentially be explained by a relatively larger sphere of DBS current distribution in mice that captures all critical fibers immediately. Finally, there are neuroanatomical differences in PFC structure between rodents and primates, where the primate PFC is more extensive. Thus, findings in rodent models cannot speak to the importance of PFC regions that lack a rodent homologue. However, directly modulating the IC in both OCD patients and in SAPAP3^{-/-} mice is effective in reducing symptoms, suggesting that human compulsivity and rodent compulsive-like behavior may share a sufficiently similar neuroanatomy.

2. Closely related, I would request that they soften/moderate the claims such as “our data outlines a specific role for mOFC neurons in the therapeutic effects of DBS in compulsive

behavior". For instance, perhaps call this a "potentially important role"? Given that we do not quite know that SAPAP grooming is "compulsive" (as opposed to, e.g., more tic-like), it would be more fair to call it "compulsion-like" or "excessive". We need to avoid anthropomorphizing the animal model.

We agree with the reviewer that anthropomorphizing animal behavior is to be avoided. Even though SAPAP3 KO mice groom to an extent that is harmful to themselves, we cannot know for sure whether this excessive grooming is truly compulsive in nature (i.e., whether mice feel the irresistible urge to perform grooming). Therefore, we removed any mention of "compulsive grooming" and replaced it with "compulsive-like grooming" or "excessive grooming". In addition, as requested by the reviewer, we softened the mOFC claim.

3. There are a few places where the argument and conclusions will be strengthened by small supplemental analyses that do not require any additional experiments:

a. They should demonstrate that there is no grooming behavior difference based on the site of GRIN lens implantation (in WT or KO), given that these do cause very large lesions that may affect behavior. Or, if it is there, they could show that it is much smaller than the effect of DBS.

The reviewer raises an interesting point. GRIN-lens implantation induces a lesion that could potentially alter grooming. To address this concern, we analyzed "baseline" grooming (during "no-DBS" periods) of implanted animals and compared it to that of surgery-naive animals that did not undergo GRIN-lens implantation. The result of this analysis demonstrates no significant effect of GRIN-lens implantation on grooming behavior, suggesting that these unilateral lesions do not alter "baseline" grooming substantially (supplementary figure 3b).

b. They never quite establish that it is the sustained-activated neurons that relate to grooming. That is, there is a bit of an intellectual jump between Figure 3/5 and Figure 6. I would like to see that correlation actually tested, e.g., to see some plot comparing "number of sustained-activated cells for DBS setting X" against "percentage reduction in grooming". Since they appear to be claiming that this is the mechanism, some correlation needs to be shown, or if it is not there, they need to say why this is reasonable, particularly because they appear to formally make the claim of that correlation at line 299-300.

To further investigate the idea that sustained-activated neurons are related to grooming reduction, we now performed a correlation analysis of the percentage of sustained-activated

neurons with the percentage reduction in grooming (per reviewer request). In support of our idea, we found a negative correlation (i.e., the more sustained-activated neurons are recruited, the more grooming is reduced) in IOFC and mOFC. This is consistent with our results indicating that mOFC might be involved in grooming reduction (supplementary figure 3i). Together with the finding that sustained-activated neurons paralleled the duration of grooming reduction and no sustained-activated neurons were observed during the anesthetized state, we believe that the correlation analysis strengthens the conclusion about an involvement of sustained-activated neurons in grooming reduction.

c. They established “locomotion associated neurons” by median splitting the speed. They really should show a distribution to verify that this is reasonable (that there is something representing bimodality and that the median is the correct quantile).

We thank the reviewer for requesting to better validate the median split of movement speed that we performed to differentiate between inactive and locomotion epochs. While re-analyzing the movement data, we found a small mistake in our Matlab code that led to an inaccuracy in figures 6a,b,c,d. We thoroughly double-checked all other figures for the same or similar mistakes and verified that this mistake did not occur anywhere else. We updated figures 6a,b,c,d. This mistake did not alter the interpretation of the data.

In order to classify the three behavioral categories, “grooming”, “locomotion”, and “inactivity, we did the following: After removing grooming data points (since grooming is a form of movement that is not locomotion), we performed a median split on the remaining data. Together, this yielded three mutually exclusive data sets: 1) movement speed during grooming, 2) non-grooming movement speed below median (“inactive”), and 3) non-grooming movement speed above median (“locomotion”). With regard to the reviewer’s request, we now conducted an additional analysis: We compared movement speed for these data sets statistically. We found movement below the median to be significantly slower than grooming, whereas movement above the median was significantly faster than grooming, suggesting that movement speed between these categories is sufficiently different to label them as different types of behavior and that it is justified to assume that animals are relatively inactive at below-median speed and that most above-median speed is linked to locomotion (supplementary figure 6b). Thus, while we agree with the reviewer that this is not the most elegant solution, we believe that it is useful.

4. They need to discuss more the finding of Figure 5d, that the populations of lower vs higher intensity do not really overlap. Yes, the finding is statistically significant, but a <20% overlap means that most neurons are discordant between current levels. This is surprising, and

deserves at least speculation to how it could arise (and needs to be flagged for the generalist reader as very surprising). Does this imply that some neurons are getting “overdosed” and no longer responding? Are they claiming some form of reverberant network effect?

The reviewer noticed that, although there is a significant overlap between neuron populations recruited at therapeutic intensities, the percentage of overlapping populations is relatively low. It is indeed surprising that the majority of DBS 300 μ A recruited neurons are in fact “newly” recruited neurons not part of the DBS 200 μ A recruited neurons. Given that we believe that our data suggests that sustained-activated neurons are driven substantially by indirect network effects, we speculate that increasing DBS parameters alters different brain networks overall, resulting in recruitment of different neuron populations. We do not believe neurons get “overdosed”, as DBS stimulation blocks (100, 200, or 300 μ A) are randomized across animals and we find similar percentage overlap across animals and regions. This point has been added to the discussion.

Revised discussion:

Moreover, to our surprise, although significantly above chance, only partially overlapping neuron populations were recruited at therapeutic intensities (i.e., consistently across brain regions). We interpret this as another indication of DBS network effects because the electrode position presumably remained stable relative to the surrounding IC axons, and thus variance in identity of recruited cells is at least partially determined by other factors than entrainment or direct disturbance of IC axonal activity.

Minor Comments:

5. The title is too strong, especially since at best, this speaks to therapeutic mechanisms of IC DBS for compulsive behavior. “Cortico striatal mechanisms of internal capsule DBS in a rodent compulsivity model” would be a good replacement.

We changed the title of the manuscript to describe our work more precisely:

“Unraveling the therapeutic mechanism of deep-brain stimulation of the internal capsule in a mouse model”

6. As per my comments in summarization above, they might consider citing some of Warren Grill’s work on informational lesions as a DBS mechanism in PD, as possibly the same concept is at play here. Ref 46 is a start but there are some broader theoretical pieces.

We thank the reviewer for suggesting to cite Warren Grill's work. We extended the introduction and discussion to better reflect how our work relates to the "information lesion" theory accordingly. In addition, we much appreciate the reviewer's observation that DBS' reduction of the number of neurons whose firing is correlated with grooming, is consistent with the "information lesion" hypothesis. This has also been added to the discussion.

7. They need to remove the "trend" claim at line 1051 and 221. There are a number of papers discussing why it is not the case that "with a few more data points this would have been significant". Having this claim detracts from an otherwise really methodologically and analytically robust paper.

We agree with the reviewer. After correcting the small mistake (see reviewer's comment 3c) and re-analyzing the data that produced figure 6a,b,c,d, we found that DBS significantly recruits "grooming-associated" neurons dose-dependently in the mOFC after all ($p=0.024$). Importantly, in all cortical regions, DBS recruited "not-associated" neurons dose-dependently, suggesting that DBS did not recruit neurons based on "cell identity", but simply recruits more neurons with higher stimulation intensities and that these happen to be "grooming-associated" in the mOFC. Thus, we believe that it is not necessary to change our conclusions.

8. It is a little surprising to see in Supp 1 that the behavior effect was not sensitive to electrode placement. One or two sentences of speculation or discussion on why this is would be nice (given the intense focus in the clinical literature on tractographic mapping and targeting). Possibly this is simply a function of how large the probe is relative to the mouse IC, i.e. the volume of activated tissue is always so large as to cover most of the fibers? It sounds like they have actually done this modeling based on the Methods.

We agree with the reviewer. This came to us as a surprise as well. Based on a comment by reviewer 1, we ran Bayesian correlations to determine whether the data support a lack of effect. Indeed, for all three experiments, we found substantial evidence ($BF>3$; current: 6.2, pulse width: 3.7; frequency: 4.9) supporting the lack of effects, which is indeed surprising given the focus of clinical literature on tractography mapping and targeting. As the reviewer suggests, we speculate that our DBS parameters in combination with the electrodes used induce a relatively large sphere or current distribution, which alters overlapping IC regions despite (slightly) different electrode coordinates in the ventral IC.

9. If they have room for extra references, they could bolster the (correct) point that DBS recruits cortical regions by referencing some of the recent human work combining cortical recording (acute and chronic) with ALIC stimulation, all of which also did show cortical changes and possibly biomarkers of compulsivity. Examples are Olsen et al. Frontiers 2019, Figue et al Nature Neurosci 2013 (cited but not in this context), Basu et al Nature BME 2021 (not exactly ALIC but close). There are of course also a number of DBS-lead recording papers including their own work.

We thank the reviewer for suggesting these publications and added them to support the claim that DBS recruits cortical regions.

10. Given that technically in the striatum they recorded inhibitory neurons which are probably but not guaranteed to be MSNs, it would probably be best to describe these as “putative medium spiny neurons”.

We agree with the reviewer and added “putative” to “medium spiny neurons”. In addition, we think this is also true for “cortical pyramidal neurons” (Yaguchi, et al., 2013), which we updated, too.

11. For the pulse width, could they please specify whether 40/80/160 us is the width of the full pulse (both phases) or solely of the cathodal phase? This causes confusion in the literature, especially since animal work tends to use symmetric biphasic whereas human DBS uses asymmetric pulses and reports only the cathodal width. (Incidentally, line 493 incorrectly gives durations in uA.)

Line 493 was corrected and information added regarding pulse width. Changes in pulse-width duration concerned both phases of the biphasic pulse equally (160us width = 80us cathodal pulse + 80us anodal phase).

We would like to express our appreciation for the time and energy the reviewer put into writing this meticulous and thorough review. This improved the manuscript considerably. We hope that we have addressed all the concerns sufficiently.

REVIEWERS' COMMENTS

Reviewer #1 (Remarks to the Author):

Authors have successfully addressed all concerns.

Reviewer #2 (Remarks to the Author):

The prior points (technically strong, well written, good N, good controls, many others) have not changed. They have made a strong response to reviewer comments, and have for the most part done a good job of moderating and explaining claims. My residual comments are minor and do not require a re-review (though I would appreciate them taking the ~10 minutes required to make these edits). I really appreciated some of the discussion added in response to another Reviewer talking about how the anesthesia results argue for a network rather than an antidromic effect.

I have only two additional requests.

1. I do think the title would fit better if they said “mechanisms” instead of “therapeutic mechanism”, i.e. remove the word “therapeutic”. These are clearly mechanisms of IC DBS. Whether they in fact are therapeutic mechanisms (and whether the singular “mechanism” exists) seems like a bit stronger claim, given everything we already discussed about what SAPAP grooming does and does not mean.

2. I respect their choice on my prior comment about locomotion, choosing to show that there is a immobility < grooming < locomotion difference in their existing categorization. On the other hand, this *is* weaker than a continuous analysis, and may still impose a categorical structure that does not fit the data. Given that this does not change the major conclusions, I do not see that any further analysis is needed. But, please add one small sentence, in the Methods or Discussion, acknowledging that small limitation.

Otherwise, my only complaint on this paper is that it is going to take hours explaining it to my lab's journal club. Very strong work.

REVIEWER COMMENTS

Reviewer #1 (Remarks to the Author):

Authors have successfully addressed all concerns.

Reviewer #2 (Remarks to the Author):

The prior points (technically strong, well written, good N, good controls, many others) have not changed. They have made a strong response to reviewer comments, and have for the most part done a good job of moderating and explaining claims. My residual comments are minor and do not require a re-review (though I would appreciate them taking the ~10 minutes required to make these edits). I really appreciated some of the discussion added in response to another Reviewer talking about how the anesthesia results argue for a network rather than an antidromic effect.

I have only two additional requests.

1. I do think the title would fit better if they said “mechanisms” instead of “therapeutic mechanism”, i.e. remove the word “therapeutic”. These are clearly mechanisms of IC DBS. Whether they in fact are therapeutic mechanisms (and whether the singular “mechanism” exists) seems like a bit stronger claim, given everything we already discussed about what SAPAP grooming does and does not mean.

We changed the title of the manuscript to describe our work more precisely:

“Unraveling the mechanisms of deep-brain stimulation of the internal capsule in a mouse model”

2. I respect their choice on my prior comment about locomotion, choosing to show that there is a immobility < grooming < locomotion difference in their existing categorization. On the other hand, this *is* weaker than a continuous analysis, and may still impose a categorical structure that does not fit the data. Given that this does not change the major conclusions, I do not see that any further analysis is needed. But, please add one small sentence, in the Methods or Discussion, acknowledging that small limitation.

We agree with the reviewer that other analyses could be informative. We have added this to the Methods (“Miniscope calcium-imaging analyses: Behavior-associated neurons”).

“Although other analyses could be informative too (e.g., continuous analysis, general linear model), we employed this analysis for simplicity.”

Otherwise, my only complaint on this paper is that it is going to take hours explaining it to my lab's journal club. Very strong work.